# ML-AMPSIT: Machine Learning-based Automated Multi-method Parameter Sensitivity and Importance analysis Tool

**Dario Di Santo**[1], **Cenlin He**[2], **Fei Chen**[3], and **Lorenzo Giovannini**[1]

[1]Department of Civil, Environmental and Mechanical Engineering, University of Trento, Trento, Italy
[2]NSF National Center for Atmospheric Research (NCAR), Boulder, CO, USA
[3]Division of Environment and Sustainability, Hong Kong University of Science and Technology, Hong Kong SAR, China

**Correspondence:** Dario Di Santo (dario.disanto@unitn.it)

**Abstract.** The accurate calibration of parameters in atmospheric and Earth system models is crucial for improving their performance but remains a challenge due to their inherent complexity, which is reflected in input–output relationships often characterised by multiple interactions between the parameters, thus hindering the use of simple sensitivity analysis methods. This paper introduces the Machine Learning-based Automated Multi-method Parameter Sensitivity and Importance analysis Tool (ML-AMPSIT), a new tool designed with the aim of providing a simple and flexible framework to estimate the sensitivity and importance of parameters in complex numerical weather prediction models. This tool leverages the strengths of multiple regression-based and probabilistic machine learning methods, including LASSO (see the list of abbreviations in Appendix B), support vector machine, classification and regression trees, random forest, extreme gradient boosting, Gaussian process regression, and Bayesian ridge regression. These regression algorithms are used to construct computationally inexpensive surrogate models to effectively predict the impact of input parameter variations on model output, thereby significantly reducing the computational burden of running high-fidelity models for sensitivity analysis. Moreover, the multi-method approach allows for a comparative analysis of the results. Through a detailed case study with the Weather Research and Forecasting (WRF) model coupled with the Noah-MP land surface model, ML-AMPSIT is demonstrated to efficiently predict the effects of varying the values of Noah-MP model parameters with a relatively small number of model runs by simulating a sea breeze circulation over an idealised flat domain. This paper points out how ML-AMPSIT can be an efficient tool for performing sensitivity and importance analysis for complex models, guiding the user through the different steps and allowing for a simplification and automatisation of the process.

## 1 Introduction

One of the primary sources of error in atmospheric and Earth system models stems from inaccurate parameter values (Clark et al., 2011; Li et al., 2018), which can affect different physical parameterisations. Although model parameter tuning can help to alleviate this issue, determining optimal values is highly dependent on model structures and how input parameters influence model outputs. Sensitivity analysis is commonly used to evaluate these input–output relationships and parameter importance, but traditional one-at-a-time (OAT) methods yield varying results depending on the interdependence of parameters, particularly within complex models, leading to issues of poor reproducibility and an inability to generalise results. Consequently, more advanced variance-based techniques like the Sobol method, in the context of global sensitivity analysis (GSA, Saltelli et al., 2008), exhibit superior performance in such tasks, albeit being computationally intensive (Herman et al., 2013) and sometimes infeasible, especially when dealing with complex weather and/or climate models like the widely used Weather Research and Forecasting (WRF) model (Skamarock et al., 2021).

An alternative approach that avoids running numerous model realisations is the utilisation of surrogate models or emulators (Queipo et al., 2005; O'Hagan, 2006; Forrester

et al., 2008; Fernández-Godino et al., 2017; Kim and Bouk­ouvala, 2020; Longo et al., 2020; Lamberti and Gorlé, 2021). A surrogate model or emulator is a simpler model trained using the input–output pairs of the original complex high-fidelity model that can be used to substitute it. The emula­tor makes the model process more computationally efficient in producing model realisations, while it still provides ac­curate predictions of the output variable. Machine learning (ML) algorithms designed for regression tasks offer a com­putationally efficient means to build surrogate models to be used for sensitivity analysis (Engelbrecht et al., 1995; Shen et al., 2008; Muthukrishnan and Rohini, 2016; Antoniadis et al., 2021; Torres, 2021; Zouhri et al., 2022). Over time, a variety of algorithms have been tested in the literature and used in different fields.

These algorithms can also be used to extract feature im­portance, which has become a well-established methodol­ogy widely employed in different geoscience fields, such as landslide susceptibility (Yilmaz, 2010; Catani et al., 2013; Pradhan, 2013; Youssef et al., 2016; Kalantar et al., 2018; Lee et al., 2018; Zhou et al., 2018; Chen et al., 2020; Liu et al., 2021; Daviran et al., 2023; Elia et al., 2023), for­est fire susceptibility (Oliveira et al., 2012; Bar Massada et al., 2013; Arpaci et al., 2014; Pourtaghi et al., 2016; Satir et al., 2016; Gigović et al., 2019), water quality assessment (Palani et al., 2008; Rodriguez-Galiano et al., 2014; Sarkar and Pandey, 2015; Haghiabi et al., 2018; Shah et al., 2021; Alqahtani et al., 2022; Trabelsi and Bel Hadj Ali, 2022), hy­drological modelling (Zhang et al., 2009; Yu et al., 2024), air quality assessment (Suárez Sánchez et al., 2011; Yu et al., 2016; Maleki et al., 2019; Sihag et al., 2019; Lei et al., 2023), groundwater mapping (Rahmati et al., 2016), agron­omy (Kok et al., 2021; Sridhara et al., 2023; Wu et al., 2023), climatological applications (Wu et al., 2021; Dey et al., 2022), renewable energy (Wolff et al., 2017; Meenal et al., 2022), and earthquake detection (Murti et al., 2022), and it also has significant relevance in civil engineering (Tian, 2013; Gholampour et al., 2017; Farooq et al., 2020; Salmasi et al., 2020), genetics (Sharma et al., 2014), biology (Cui and Wang, 2016), and medical research (Antonogeorgos et al., 2009; Maroco et al., 2011; Yang et al., 2022).

ML techniques have gained traction in weather and cli­mate modelling and observations (Schultz et al., 2021; Schneider et al., 2022), particularly in parameter optimisa­tion tasks like calibration (Bocquet et al., 2020; Bonavita and Laloyaux, 2020; Williamson et al., 2013; Couvreux et al., 2021; Dagon et al., 2020; Watson-Parris et al., 2021; Cin­quegrana et al., 2023), spatial interpolation (Stein, 1999; Sekulić et al., 2020), downscaling (Fowler et al., 2007; Ma­raun and Widmann, 2018; Leinonen et al., 2021), parame­terisation substitution (Rasp et al., 2018; Han et al., 2020; Yuval and O'Gorman, 2020; Mooers et al., 2021; Grundner et al., 2022; Ross et al., 2023), and image-based classification (Chase et al., 2022, 2023).

Among the most relevant for the topic of the present work, Dagon et al. (2020) focused on building a surrogate model based on feed-forward artificial neural networks of a land surface model (CLM5) ensemble of perturbed parameters, which greatly improved the rapidity of generating predic­tions. Similarly, Cinquegrana et al. (2023) built a frame­work for optimising physical parameters for the Icosahedral Nonhydrostatic (ICON) limited area model at a high resolu­tion, aiming to reduce the discrepancy between observed and modelled meteorological variables using an efficient global optimisation algorithm relying on Gaussian-based surrogate models. Watson-Parris et al. (2021) introduced an open-source tool (ESEM) based on surrogate models for model calibration and uncertainty quantification, demonstrating its functionalities for climate modelling. Couvreux et al. (2021) used Gaussian process-based methods to calibrate parame­ters through the comparison of single-column simulations and reference large-eddy simulations over multiple boundary layer cases.

Despite these recent advancements, the extraction of fea­ture importance remains relatively uncommon in the meteo­rological and/or climate modelling literature. A notable ex­ception in recent times is Baki et al. (2022), who employed GSA methods and a surrogate model based on Gaussian pro­cess regression. The study found that a subset of parame­ters significantly influenced WRF simulations of tropical cy­clones over the Bay of Bengal. Similarly, Fischer et al. (2024) used a Gaussian process regression-based surrogate model of the ICON model to quantify the uncertainty of simulations of the African monsoon through GSA.

Many of the previous studies have proposed comparisons between several feature importance analysis algorithms. This is because the ability of these algorithms to best capture fea­ture relevance is influenced by a variety of factors that can change with the application, depending on the context under analysis, such as the degree of non-linearity of the input–output relationships, the interaction degree between features, the dimensionality of the features, the size and quality of the data used for training, the shape and smoothness of the dis­tribution of the training data, and ultimately the validity of each algorithm's assumptions. Often, the influence of one or more factors on the chosen method's quality cannot be as­sessed in advance, leading to a trial-and-error procedure that would benefit from a multi-algorithm approach where the re­sults of different methods can be compared. For this reason, the present work shares the same multi-method philosophy as many of the studies mentioned above, extracting the most popular algorithms available in the literature and combining them into a single flexible, efficient framework for analysis.

The importance of sensitivity analysis in Earth science modelling is critical not only for academic pursuits but also for its practical implications for public safety and re­source management. Currently, the diversity of research cul­tures across scientific disciplines, coupled with heteroge­neous computational resources and varying degrees of fa-

miliarity with sensitivity analysis techniques, contributes to a predominant reliance on older, more familiar methods. This scenario prevails despite the increasing complexity of models, which would require more robust sensitivity analysis techniques. The field of meteorology currently exhibits a significant gap in the adoption of advanced sensitivity analysis methods despite the chaotic nature of atmospheric dynamics, and the interactions among numerous parameterisations in atmospheric models contribute to a high degree of sensitivity to input parameter variations, underscoring the need for robust uncertainty quantification to improve model reliability.

In light of the above considerations and to fill this gap, this paper proposes a new tool, the Machine Learning-based Automated Multi-method Parameter Sensitivity and Importance analysis Tool (ML-AMPSIT), which aims to provide a flexible and easy-to-use framework for performing sensitivity and importance analysis for complex models. ML-AMPSIT applies a series of ML feature importance extraction algorithms to model parameters (using the widely used WRF/Noah-MP coupled meteorological model as a case study), accommodating any user-specified model configuration. ML-AMPSIT represents a novel contribution to the field by providing a toolkit that integrates multiple ML algorithms for an improved sensitivity analysis. The algorithms included are among the most commonly used in the literature, namely LASSO (see the list of abbreviations in Appendix B), support vector machine, classification and regression trees, random forest, extreme gradient boosting, Gaussian process regression, and Bayesian ridge regression. These algorithms have been chosen for their simplicity and speed and to create an ensemble of state-of-the-art ML models, each employing distinct methodologies, so as to improve the flexibility of the tool and its performance in different possible applications. This diversity allows for a robust method of self-validation or self-falsification of the results through comparative analysis, enhancing the reliability of the findings by ensuring that consistent results are not an artefact of a single modelling approach. While most of these algorithms directly provide a measure of feature importance through the Python scikit-learn library, the last two methods are specifically used in this framework for a fast implementation of the Sobol method through the SALib Python library, leading to a computationally efficient way to obtain the Sobol sensitivity indices directly from the ML-inferred relation between input and output data. Our tool's objective is, overall, to assist users in evaluating parameter sensitivity and importance using computationally inexpensive and non-linear interaction-aware approaches.

ML-AMPSIT guides the user through the different steps of the sensitivity and importance analysis, allowing, on the one hand, for a simplification and automatisation of the process and, on the other hand, for an extension of the application of advanced sensitivity and importance analysis techniques to complex models through the use of computationally inexpensive and non-linear interaction-aware methods. Once the user knows which parameters cause most of the variance within a perturbed ensemble, the user can potentially concentrate on these parameters to improve model results. Indeed, knowing which parameters are most critical to the simulation output highlights which values should be estimated with more care to improve model results.

This paper is organised as follows: Sect. 2 outlines the methodology used to develop ML-AMPSIT, including a detailed description of the ML models integrated into the tool and the workflow for performing sensitivity and importance analysis. Section 3 presents the case study involving the coupled WRF/Noah-MP model to demonstrate the application of ML-AMPSIT. The results of the sensitivity analysis are discussed in Sect. 4, highlighting the effectiveness of different ML models in identifying the key parameters for the case study presented in this paper. Finally, Sect. 5 concludes the paper with a summary of the findings and some insights into potential future work to further enhance the capabilities of ML-AMPSIT.

## 2 Methods

In this section, we describe the methodological framework underlying this study. We begin with an overview of the ML-AMPSIT workflow, detailing the process from the selection of the input parameters to the sensitivity analysis phase. We then introduce the Sobol method, a variance-based technique used for GSA. Finally, we provide a description of the ML algorithms integrated into the tool, highlighting their main characteristics, how they are implemented and used in ML-AMPSIT, and the rationale behind their selection.

### 2.1 ML-AMPSIT workflow

The ML-AMPSIT workflow (Fig. 1) can be divided into four main steps, each of which involves one or more Python- or Bash-based scripts: the pre-processing phase, the model run phase, the post-processing phase, and the sensitivity analysis phase.

1. The selection of the input features is accomplished by specifying the parameter names within the configuration file configAMPSIT.json. The compiled configuration file related to the case study discussed in this paper is reported in the Appendix. There is no upper limit for the number of parameters that can be analysed, but it is worth noting that the sensitivity analysis could converge significantly more slowly in high-dimensional (i.e. with more parameters) problems. Moreover, the scalability with the number of parameters can depend strongly on the case study considered. The number of simulations to be performed must also be specified through the configuration variable totalsim. To generate the values of the parameters to be tested, a Sobol sequence of the same length as totalsim is produced for each parameter

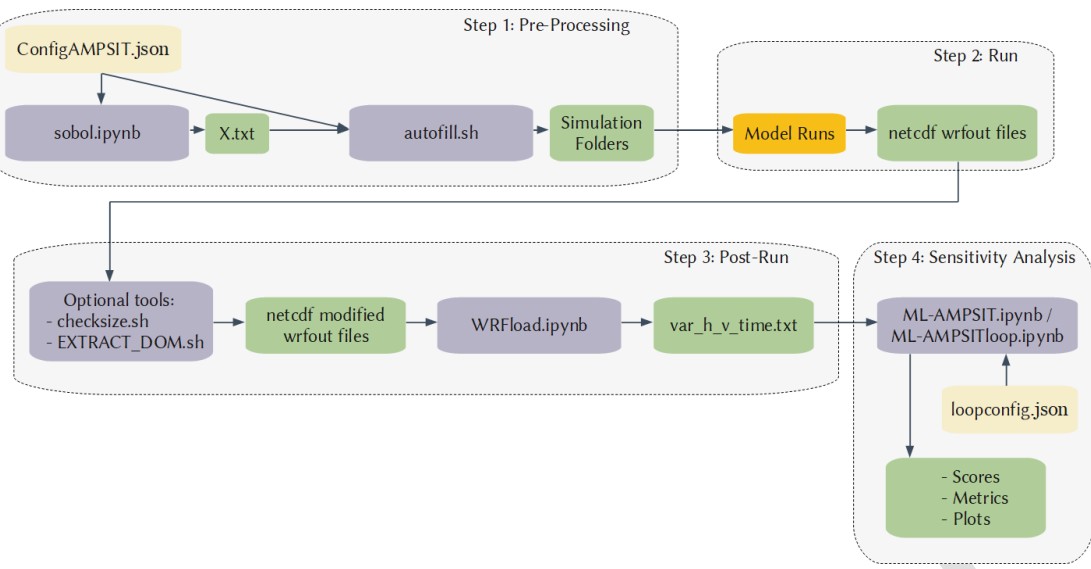

**Figure 1.** ML-AMPSIT workflow. The main code scripts are indicated with blue boxes. The yellow boxes indicate the configuration files that need to be filled by the user, the green boxes refer to the output files that eventually become inputs for other subsequent scripts, and the orange box indicates the generic model execution that varies depending on the model involved.

from the pre-processing script sobol.ipynb. The Sobol sequence (Saltelli et al., 2010; Bratley and Fox, 1988) is a quasi-random low-discrepancy distribution designed to produce well-spaced points in the unit hypercube representing the parameter space. Unlike random sampling, each point in the sequence considers the positions of the previous points, resulting in a more uniform filling of gaps, as shown in Fig. 2. Consequently, a robust sequence is generated more efficiently compared to random sampling, requiring fewer points.

Once the Sobol sequences are generated, a user-specified reference value and a maximum perturbation percentage need to be specified in configAMPSIT.json, which will be passed to the preprocessing script autofill.sh. These values are used to rescale the sequence values from the $[0, 1]$ range to the actual parameter range space. The output of the sobol.ipynb script is the file X.txt, containing a $m \times n$ matrix, where $m$ is the number of simulations, and $n$ is the number of parameters tested.

Therefore, each row specifies a different set of parameter values that will be used in each particular model realisation. Based on these data, the autofill.sh script creates multiple copies of the folder in which the model is run and then searches for each parameter name within the original model parameter look-up table in each newly created folder. The values of the parameters are then changed according to the X.txt file for each realisation. Since this script edits the original model parameter look-up table, which is MPTABLE.TBL for the WRF/Noah-MP model in the case study presented here,

it is necessarily model-dependent and thus needs to be adapted if used with other models to suitably modify the values of the tested parameters.

2. After all the simulation folders have been created, the user can run the original high-fidelity model as usual. It will be necessary to collect all the output files into one single folder, whose path must be specified in the configuration file, so that the post-processing script can find it.

3. Once the user has completed all the high-fidelity model runs, a post-processing script named WRFload.ipynb is provided to extract single-point time series for each output variable at specific coordinates in the simulation domain, as specified by the user in the configuration file. The resulting output data, which serve as input to the sensitivity analysis tool ML-AMPSIT.ipynb, consist of different files with the naming convention var_h_v_time.txt, where "var" is the variable name, $h$ and $v$ indicate the labels identifying the horizontal and vertical grid cells, while "time" represents the simulated time. The script WRFload.ipynb specifically extracts variables from NetCDF files that follow a WRF-like format (a widely used format for weather and climate models). If the user's model output follows a different format, the script must be modified accordingly.

Extracting single-point time series means that the tool has no information about spatial patterns and it cannot capture time patterns because the sensitivity analysis is performed on each single point and time separately. Consequently, the importance of different parameters

can be directly compared only for the specific point and time being analysed. However, the observation of consistent relative importance between parameters across different points and times can confirm the statistical robustness of the results.

4. The sensitivity phase is performed by the main script of the ML-AMPSIT tool, ML-AMPSIT.ipynb, which accomplishes a regression task based on different ML algorithms offered to the user. As mentioned in the Introduction, this multi-method approach is useful for comparing different results since each algorithm is structurally different and could be more or less appropriate for the problem at hand. This ML-based ensemble philosophy is similar to an ensemble learning (EL) approach (Ren et al., 2016), which combines the predictions of multiple base models to improve the overall performance, but the present tool does not yet include an option to integrate the methods into a stacking, bagging, or boosting procedure, allowing for the user to choose any single method or multiple methods independently. For each method offered by the present tool, the input and output data are split into training and testing sets in proportions of 70 % and 30 %, and each set is scaled separately to have zero mean and unit variance with respect to the ensemble. The training set is used to fit the model to the data, while the testing set is used to evaluate the model's ability to reproduce new data. This strategy is used to mitigate the risk of overfitting. The coefficient of determination ($R^2$), the mean squared error (MSE), and the mean absolute error (MAE) are used as measures of goodness when comparing the predicted output against the actual "truth", i.e. the results of the original high-fiedlity model simulations. Since all the variables are scaled before calculating these error metrics, MSE and MAE are not affected by the different scales of the variables. This allows for a fair and meaningful comparison of the model's performance across different variables. The coefficient of determination $R^2 = 1 - \frac{\text{SS}_{\text{res}}}{\text{SS}_{\text{tot}}}$ is used as a measure of goodness of fit, where $\text{SS}_{\text{res}}$ is the residual sum of squares, and $\text{SS}_{\text{tot}}$ is the total sum of squares. $R^2$ indicates how much variation in the target variable can be explained by the model's predictors. $R^2$ is typically a value between 0 and 1, where values closer to 1 indicate a better ability of the model to explain the variance in the data. Eventually, if the chosen model fits worse than the average value then $\frac{\text{SS}_{\text{res}}}{\text{SS}_{\text{tot}}}$ can be greater than 1, and $R^2$ is negative. If the model has low values of MSE and MAE but also low values of $R^2$, it might indicate that the relationship between the input data and the target variable cannot be properly explained in terms of linear weights only. This is an indication of non-linearity in the output response. In addition to the $R^2$ coefficient, the associated $p$ value is also computed and saved.

The script ML-AMPSIT.ipynb produces an interactive graphical user interface (GUI) built from the ipywidgets Python library, which allows the user to specify which vertical level and surface point to consider in the analysis, the output variable for which to compute the sensitivity analysis, the number of simulations to consider, the algorithm to use, and the output time to plot for punctual evaluations. The flexibility of this GUI allows the user to quickly check the influence of the number of simulations on the robustness of the results and the performance of the different ML methods implemented.

Based on the selected options, unless the specified methods are Gaussian process regression or Bayesian ridge regression, the tool produces four plots: the upper two are dedicated to the feature importance time series and the time evolution of the metrics for the whole simulation duration, and the lower two show the metrics and feature ranking specific to a particular time selected. Hence, the user is provided with both the global result and the analysis related to a single output time. An example from the proposed case study will be provided in Sect. 3.3. If the specified method is either Gaussian process regression or Bayesian ridge regression, the features are ranked based on the total Sobol index, and the tool produces two additional plots, one showing the second-order Sobol interaction index between each couple of parameters and the other showing the feature ranking based on the first-order Sobol index, which could potentially be different from the total index-based ranking if the parameters' interactions are strong enough.

An alternative version of the main script is the ML-AMPSITloop.ipynb loop suite, which allows the user to automate multiple analyses by using the loopconfig.json configuration file, specifying all the combinations of settings to be explored without the need to manually set each combination through the graphical interface, saving time for long in-depth analyses.

## 2.2 Sobol method

In classical sensitivity analysis, given a set of input parameters $\{X_1, .. X_k\}$, the elementary effect of a single perturbation $\Delta$ in the input parameter $X_i$ on the output $Y(X)$ is defined as (Saltelli, 2007):

$$\text{EE}_i = \frac{[Y(X_1, X_2, \ldots, X_i + \Delta, \ldots, X_k) - Y(X_1, X_2, \ldots, X_i, \ldots, X_k)]}{\Delta}. \quad (1)$$

The above definition assumes a linear relationship between the parameter and the output variable, and it becomes ineffective in the presence of non-linearities or interactions between parameters. To achieve the highest level of generalisation in

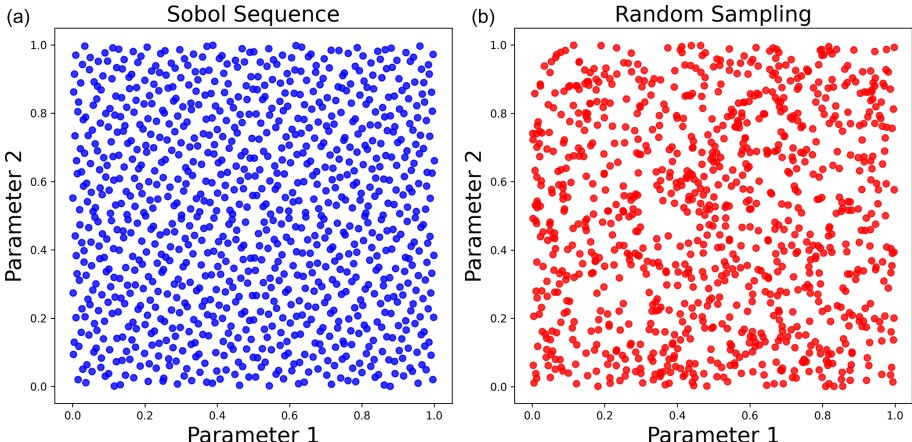

**Figure 2.** Demonstration of the differences between Sobol sampling (**a**, blue dots) and random sampling (**b**, red dots) in representing the parameter space. The Sobol sequence is able to more uniformly cover the parameter space, avoiding the presence of very close points, as it occurs in the random sequence.

sensitivity analysis, both the effect of the single input parameter and the additional effect of its interaction with other parameters must be evaluated.

A variance-based approach achieves this while also not relying on a linear assumption. The most well-established variance-based method is the Sobol method (Saltelli and Sobol', 1995). With this approach, the variance $V(Y)$ is decomposed as follows:

$$V(Y) = \sum_i V_i + \sum_i \sum_{j>i} V_{ij} + \ldots + V_{12\ldots k}, \quad (2)$$

where $V_i$ is the main effect variance, representing the contribution of the $i$th input parameter to the output variance, and $V_{ij}$ is the second-order interaction effect variance, representing the combined contribution of the $i$th and $j$th input parameters to the output variance and so on up to $V_{12..k}$, which represents the interaction effect variance of all $k$ input parameters together.

Dividing by $V(Y)$, the Sobol indices are derived as $S_i = V_i/V(Y)$, $S_{ij} = V_{ij}/V(Y)\ldots$, leading to

$$1 = \sum_i S_i + \sum_i \sum_{j>i} S_{ij} + \ldots + S_{12\ldots k}. \quad (3)$$

The total effect or total index $S_{\mathrm{Ti}}$ is the total contribution to the output variation due to a specific factor $X_i$, i.e. a specific parameter. Hence, for instance, for a set of three parameters $X_1, X_2, X_3$, the total effect index for the parameter $X_1$ is

$$S_{\mathrm{T1}} = S_1 + S_{12} + S_{13} + S_{123}. \quad (4)$$

Both first-order effects $S_i$ and total effects $S_{\mathrm{Ti}}$ are important to assess the overall influence of an input parameter.

## 2.3   Implemented ML algorithms

### 2.3.1   LASSO

The least absolute shrinkage and selection operator (LASSO, Tibshirani, 1996) is an ML method used for feature selection and regression. It is derived from the basic concept of curve fitting in the context of optimisation; therefore, it is one of the simplest algorithms among the most widely used. The goal of LASSO is to identify a subset of input features that are most predictive of the output variable while also performing regularisation to prevent overfitting.

In a classical regression problem, the goal is to find a function that maps the input features to the output variable. This is done by minimising an objective function that takes into account the differences between the observations and the predictions as a measure of how well the model fits the data. The minimisation is performed during the training process to find the optimal values of the model coefficients. In the LASSO algorithm, a penalty term (the "regularisation") is added to the objective function, encouraging the model coefficients to be small. Specifically, the objective function for LASSO regression is the residual sum of squares (RSS), while the penalty term is based on the sum of the absolute values (L1 regularisation) of the coefficients, which promotes sparsity of the solution, and it can be adjusted to control the amount of shrinkage applied to the coefficients. As a result, LASSO can be particularly useful in identifying the most important features by setting the coefficients of the less important features to zero.

The present tool uses LassoCV from the Python library scikit-learn, which adds a cross-validation strategy to the standard LASSO algorithm. In cross-validation, data are divided into multiple subsets or "folds", and the model is trained and evaluated multiple times, each time using a different fold as the validation set and the rest of the data as

the training set (Brunton and Kutz, 2019). By averaging the results of the multiple evaluations, cross-validation provides a more accurate estimate of the model's performance than a single evaluation on a single training–validation split.

### 2.3.2 Support vector machine

Support vector machine (SVM) is an ML method that can be used for regression and classification tasks. The original algorithm was proposed in 1963 (Vapnik and Chervonenkis, 1963), was refined multiple times (Boser et al., 1992; Cortes and Vapnik, 1995; Schölkopf and Smola, 2002), and is considered to be a core method in ML.

For regression tasks, SVM aims to find a hyperplane that approximates the underlying relationship between the input variables (i.e. the model parameters) and the continuous output values. In order to do this, SVM solves a convex optimisation problem by minimising a cost function that incorporates a margin of error and an L2-norm-based regularisation term (ridge-type regularisation). Unlike the LASSO-type regularisation, the L2 regularisation is a penalty based on the square of the coefficients. This penalty term is less strict compared to LASSO because it helps to shrink the coefficient values towards zero without eliminating them completely. This can be preferred to a LASSO regularisation when there are many important parameters since it avoids eliminating them in favour of the most important ones. The regularisation term controls the trade-off between the complexity of the model and the amount of error allowed, and its strength is regulated by the hyperparameter "C". The optimisation problem in the proposed implementation is solved by the default quadratic "liblinear" Python solver. SVM offers several advantages, including the ability to handle high-dimensional data and resistance to overfitting when properly regularised. The tool presented in this paper implements a simple linear kernel, which maps the input data into a high-dimensional space using a linear function.

### 2.3.3 Classification and regression trees

The classification and regression trees (CART, Breiman et al., 1984) algorithm is one of the most basic and straightforward decision tree methods, making it widely used and popular for various applications. The main idea behind CART is to partition the input space recursively into smaller regions based on the values of the input variables, with the goal of minimising the impurity (or variance) within each resulting region. This partitioning process creates a binary tree structure called a "decision tree", where each internal node represents a splitting rule based on a selected input variable and a threshold value. The leaf nodes of the tree represent the final prediction or class assignment.

In the proposed tool, from the training dataset consisting of input–output pairs, the algorithm recursively selects the best splitting rule at each internal node based on the mean squared error of the regression. The splitting process continues until a stopping criterion is met, such as a maximum tree depth or a minimum number of samples required to split a node. For regression tasks, the predicted output is the average value of the training samples within the leaf node.

### 2.3.4 Random forest

Random Forest (RF, Breiman, 2001) is technically an ensemble learning method that can be used for both classification and regression tasks. Differently from the CART algorithm, a multitude of decision trees are constructed in an RF, and the final prediction is based on the average of the predictions of all the trees. For this reason, RF usually performs significantly better than CART.

The advantage of using RF is that it can handle high-dimensional data and complex interactions between variables, making it a useful tool for sensitivity analysis. However, it can suffer from overfitting, where the model performs well on the training data but poorly on new data. To mitigate this, both cross-validation and Bayesian optimisation are used to make the model more robust.

### 2.3.5 Extreme gradient boosting

Extreme gradient boosting (XGBoost, Chen and Guestrin, 2016) builds an ensemble of decision trees, similarly to RF, but each new tree is trained to correct the errors of the previous trees. It is hence a refined version of an RF, and it is usually expected to perform better.

### 2.3.6 Gaussian process regression

Gaussian process regression (GPR), better known in geosciences as kriging (Stein, 1999) when applied to spatial interpolation, is a non-parametric algorithm for computing the probability density function of the regression curve instead of a single fitting curve and can be used as a supervised ML technique (Rasmussen and Williams, 2005). GPR assumes that the output values follow a Gaussian distribution with an unknown mean $\mu$ and variance $\sigma$ that must be predicted given a set of input–output pairs. To achieve this, GPR models the output values as a function of the input variables, where the function is assumed to be smooth and continuous. GPR is often described as a non-parametric method because it does not assume a specific functional form for the relationship between input and output variables. Instead, it models this relationship as a distribution over possible functions, allowing for flexibility in the shape of the regression curve. However, it is important to note that there are underlying assumptions about the functional form embedded in the chosen kernel. The kernel influences the shape and properties of the functions that the Gaussian process can learn. The kernel implemented inside ML-AMPSIT is the radial basis function (RBF) kernel, which uses functions of the type

$\sigma^2$ exp $\left(-\frac{(t-t')^2}{2l^2}\right)$, ~~where $\sigma^2$~~ [TS1] ~~is a variance,~~ $l$ is a length scale, and $t$ and $t'$ represent pairs of values extracted from the training data. The choice of an RBF kernel is particularly advantageous due to its ability to model complex, non-linear relationships without imposing strong parametric constraints. The RBF kernel's smoothness assumption is well-suited for many real-world applications where the underlying function is expected to be continuous and differentiable. During the training phase, GPR estimates the parameters of this kernel function and calculates the covariance matrix between the input–output pairs. Using this covariance matrix and the training data, GPR then estimates the mean and variance of the distribution for each new input value through Bayesian inference.

### 2.3.7  Bayesian ridge regression

In Bayesian ridge regression (BRR), the hyperparameters of a classical ridge regression (i.e. a linear regression that implements a ridge regularisation term) are associated with a priorly assumed probability distribution (also called hyperprior) and tuned through training in a Bayesian inference approach (Box and Tiao, 1992). Defining both a prior distribution $p(H)$ for the model parameters $H$ and a likelihood function $p(E|H)$ for the ingested data $E$, the BRR model computes the posterior distribution over functions $p(H|E)$ given the observed data through the use of Bayes' theorem $p(H|E) = \frac{p(E|H)\cdot p(H)}{p(E)}$, where $p(E) = \int p(E|H)\cdot p(H)\,\mathrm{d}H$ is the marginal likelihood. Once the posterior distribution is obtained, the model is used to make predictions for unseen data points. These predictions come with uncertainty estimates, which are derived from the posterior distribution. BRR, as in the case of the SVM algorithm, employs an L2 regularisation; hence, it spreads the coefficient values more evenly, stabilising the model and preventing overly large coefficient estimates.

### 2.3.8  Feature importance computation

Each algorithm implemented in this study provides a method for calculating feature importance, albeit through different approaches. In principle, a single sensitivity method could be used to evaluate feature importance across all algorithms. However, some algorithms have built-in methods specifically designed to align with their inherent characteristics.

- *Fitting methods.* LASSO and SVM derive feature importance from the model coefficients. In these linear models, the magnitude of the coefficients indicates the strength and direction of the relationship between each feature and the target variable. Specifically, in the scikit-learn library, this can be accessed through the best_estimator_.coef_ attribute. Larger absolute values of these coefficients indicate greater importance.

- *Tree-based algorithms.* For CART, RF, and XGBoost, feature importance is assessed using the mean decrease in impurity (MDI) method. This method quantifies the contribution of each feature to the overall prediction accuracy by measuring how much each feature decreases the impurity of the splits in which it is involved. For RF and XGBoost, the final value is obtained by averaging over all the trees in the ensemble. In scikit-learn, these contributions are accessible through the feature_importances_ attribute. The MDI method is particularly effective because it directly measures the impact of each feature on the model's decision process, providing a clear indication of feature importance.

- *Probabilistic methods.* GPR and BRR do not have a built-in mechanism for directly assessing feature importance. Therefore, in this work, the Sobol method is used to infer feature importance. Once built and tested against the original model outputs, the GPR and BRR surrogate models can be used to perform a GSA in substitution of the original model. By using a surrogate model, the computational cost of running the original model for a large number of input combinations is avoided. Instead, the surrogate model can be used to generate a large number of input combinations with significantly less computational time and to evaluate their impact on the output. Over these samples, in ML-AMPSIT, the Sobol sensitivity indices are computed following the definition proposed by Saltelli et al. (2008). The user can then compare the Sobol indices evaluated with both GPR and BRR, providing information on their robustness and reliability. In the proposed tool, after the algorithm generates the optimal surrogate model, it uses the Python library SALib to compute the Sobol total index as a score for the sensitivity importance of each parameter. The low computational cost of these emulators allowed us to employ a surrogate sampling generated by sobol.sample(), with 5000 input values (the user can change this value by modifying the configuration parameter Nsobol in loopconfig.ipynb), with the overall Sobol method calculations performed in minutes against a single traditional WRF simulation typically taking several hours. The Sobol first-order index and Sobol second-order interaction terms are also available for users who wish to examine the presence of strong parameter interactions.

Despite the differences in the feature importance calculation approaches of the different algorithms, each method is applied to standardised, non-dimensional data, and each feature importance set is scaled between [0,1]. This ensures that feature importance scores are comparable across all the models. In general, the sum of the total Sobol indexes ($\sum S_{\text{Ti}}$) is $\geq 1$ because higher-order terms, representing the interactions between parameters, are double-counted in $\sum S_{\text{Ti}}$ (see Eqs. 3 and 4). However, it was observed that, in the present case

study, higher-order terms are mostly negligible (order $10^{-3}$), which implies that $\sum S_{Ti} \approx \sum S_i \approx 1$. Therefore, the $S_{Ti}$ indices are left non-normalised since they are still comparable with reasonable accuracy to the other normalised sensitivity indices and, at the same time, allow the evaluation of possible spurious results in the Sobol index calculation due to poor sample statistics in the convergence analysis, as will be highlighted in Sect. 4.3. For the general case, the published tool automatically scales the $S_{Ti}$ indices. The user who wants to check non-normalised values (e.g. to evaluate the effects of the interaction terms on the Sobol total indices) can find and uncomment the lines #importance_list.append(importances).

The primary objective of all these methods is to quantify the sensitivity of the model output to changes in the input features. Consequently, the feature importance scores obtained from these different methods provide a well-posed comparison of parameter sensitivities. By evaluating and comparing these scores, it is possible to gain a comprehensive understanding of the relative importance of each feature across different modelling approaches, which increases the robustness of the results.

### 2.3.9 Hyperparameter tuning

In the proposed tool, the hyperparameters of each implemented algorithm are tuned based on a cross-validation score obtained through Bayesian optimisation. Bayesian optimisation is an iterative process that seeks to explore the hyperparameter space while also exploiting regions of the space that are expected to yield good performance. At each iteration, the method proposes a new set of hyperparameters based on a probabilistic model of the function behaviour and then evaluates the function at that point. The results of the evaluation are used to update the probabilistic model, which is then used to propose a new set of hyperparameters for the next iteration.

To conclude this Methods section, Table 1 provides a summary of the main characteristics of the ML models used in this study. This table outlines the regression type of each model (linear or non-linear), the associated method used to evaluate feature importance, and other relevant characteristics such as regularisation techniques and kernel functions.

### 3 Case study

The proposed case study, adopted to highlight the functionalities of ML-AMPSIT, is based on idealised coupled WRF/Noah-MP simulations of a sea breeze circulation over a flat three-dimensional domain. The objective of this case study is to evaluate the impact of a prescribed set of Noah-MP parameters on the development of the thermally driven wind.

### 3.1 The WRF/Noah-MP model

The Weather Research and Forecasting (WRF) model is a widely used state-of-the-art mesoscale numerical weather prediction model for atmospheric research and operational forecasting applications, which is supported by the NSF National Center for Atmospheric Research (NCAR), with more than 50 000 registered users from more than 160 countries (Skamarock et al., 2021). It offers a wide range of customisation options consisting of dedicated modules and physics schemes to meet the state of the art in atmospheric science and to adapt to a wide variety of scenarios. The dynamical core used for this case study is the Advanced Research WRF (ARW), which uses a third-order Runge–Kutta scheme for time integration with a time split method for solving acoustic modes (Wicker and Skamarock, 2002) and an Arakawa-C grid staggering for spatial discretisation. In the case study presented here to illustrate the functionalities of ML-AMPSIT, WRF is coupled with the Noah-Multiparameterization (Noah-MP; He et al., 2023; Niu et al., 2011; Yang et al., 2011) land surface model (LSM). Noah-MP is one of the most used LSMs available in WRF to calculate surface–atmosphere exchanges and interactions. It is an augmented version of the Noah land surface model (Ek et al., 2003) that allows the usage of different physical schemes and multi-parameterisation options, reaching a total number of 4584 possible combinations (https://www.jsg.utexas.edu/noah-mp/, last access: 1 March 2024).

### 3.2 Model setup

Simulations are performed using one domain with $201 \times 201$ cells in the horizontal plane, with a grid spacing of 3 km. A total of 65 vertical levels are used, transitioning from a vertical resolution of 7 m close to the surface up to 600 m at the top of the domain, which is placed at 16 km above sea level. The domain is completely flat and subdivided into two equally sized rectangular sub-regions of land and water, with the interface line oriented along the west–east direction (Fig. 3). The aim is to simulate the daily cycle of a sea–land breeze. Boundary conditions are set as open at all the boundaries.

The initial atmospheric potential temperature profile is set using the following expression, representative of a stable atmosphere:

$$\theta(z) = \theta_s + \Gamma z + \Delta\theta(1 - e^{-\beta z}), \tag{5}$$

where the surface temperature $\theta_s = 280$ K, $\Gamma = 3.2$ K km$^{-1}$, $\Delta\theta = 5$ K, and $\beta = 0.002$ m$^{-1}$. The atmosphere is initially at rest, and the relative humidity is set to be constant over the entirety of the domain and equal to 30 %. The sea temperature is set as 293 K.

The idealised simulations start at 13:00 UTC on 19 March, are centred at 47° N and 11° E, and last for 35 h and thus are characterised by a solar radiation cycle representative of

**Table 1.** Summary table of the characteristics of the ML models implemented in ML-AMPSIT.

| Model name | Regression type | Feature importance method | Additional characteristics |
| --- | --- | --- | --- |
| LASSO | Linear | Model coefficients | Applies L1 regularisation to encourage sparsity in coefficients |
| SVM | Linear | Model coefficients | Uses L2 regularisation; implemented with a linear kernel |
| CART | Non-linear | Mean decrease in impurity (MDI) | Constructs binary trees for decision-making based on feature values |
| RF | Non-linear | Mean decrease in impurity (MDI) | Ensemble method that builds multiple decision trees |
| XGBoost | Non-linear | Mean decrease in impurity (MDI) | Gradient-boosting technique, correcting errors of previous trees |
| GPR | Non-linear | Sobol indices (external method) | Models output as a Gaussian distribution; uses radial basis function kernel |
| BRR | Linear | Sobol indices (external method) | Incorporates Bayesian inference with ridge (L2) regularisation |

the equinoxes at mid-latitudes. The first 11 simulation hours are not analysed and are considered to be the spin-up period. Therefore, analyses concentrate on a full diurnal cycle, from 00:00 UTC on 20 March to 00:00 UTC on 21 March. Model output is saved every hour.

The physical parameterisation schemes selected for the present work are the rapid radiative transfer model (RRTM) for longwave radiation (Mlawer et al., 1997), the Dudhia scheme (Dudhia, 1989) for shortwave radiation, and the YSU scheme (Hong et al., 2006) as the planetary boundary layer (PBL) parameterisation, coupled to the MM5 similarity scheme for the surface layer. Since this idealised study aims to reproduce a sea–land circulation, which best develops under completely clear-sky conditions, the microphysics parameterisation is switched off, along with the convective scheme, since convection is explicitly resolved at the resolution used.

As said above, the Noah-MP model is used to evaluate land–atmosphere exchange. In Noah-MP, the canopy radiative transfer scheme used is the "modified two-stream" (Niu and Yang, 2004), one of the most used, which aggregates cloudy leaves into evenly distributed tree crowns with gaps. The gaps are computed according to the specified vegetation fraction. The Ball–Berry scheme, the most common choice in the literature, is used for the stomatal resistance computation, with the Noah-type soil moisture factor (Schaake et al., 1996), while the surface runoff parameterisation TOPMODEL (Niu et al., 2007), with the groundwater option, is used for runoff and groundwater processes. The surface resistance to evaporation and sublimation processes is set following Sakaguchi and Zeng (2009). The surface layer drag coefficient, used to compute heat and momentum exchange coefficients, is calculated with the traditional Monin–Obukhov-based parameterisation. In this work, the dynamic vegetation option is not activated, with the prescription of a fixed vegetation fraction of 60 % to consider a reasonably realistic percentage, while the monthly satellite-based climatological leaf area index is read from the MPTABLE.TBL file, which contains all the parameter values. Finally, crop and irrigation options are deactivated. It is important to underline that water physical properties are not varied in the sensitivity simulations, but changes in atmospheric variables are also expected over water due to the indirect effects of the variations in the surface parameters over land.

In order to simplify this demonstrative case study, only six of the surface parameters defined in the look-up table MPTABLE.TBL, from which WRF reads and sets the surface values accordingly, are considered. In particular, the Noah-MP reference vegetation type adopted over land is grassland (vegtype=10 of the IGBP-Modified MODIS classification), and the parameters for which the sensitivity and relative importance are evaluated are the characteristic leaf dimension (DLEAF), the height of the vegetative canopy top (HVT), the momentum roughness length (Z0MVT), the near-infrared leaf reflectance (RHOL_NIR), the empirical canopy wind parameter (CWPVT), and the leaf area index for the month of March (LAI_MAR). The choice of these parameters is based on their importance in other sensitivity studies reported in the literature (Mendoza et al., 2015; Cuntz et al., 2016; Arsenault et al., 2018). The final perturbed model parameter ensemble contains 100 samples, each with different parameter values based on the associated Sobol sequences. The input ensemble is generated by perturbing the parameters by up to 50 % of their reference value in the look-up table MPTABLE.TBL. It should be clear that the results of a sensitivity analysis, regardless of the approach chosen, always depend on the range of exploration of the parameters and that their transferability to arbitrary ranges of values is not guaranteed if the true sen-

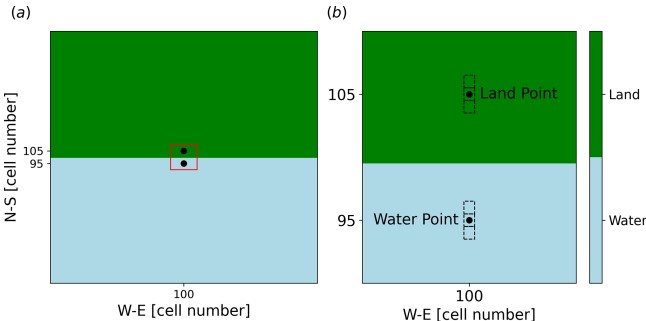

**Figure 3. (a)** Domain configuration for the idealised simulation study. The grid consists of $201 \times 201$ cells with a 3 km spacing, divided equally into land (upper region, green) and water (lower region, light blue). The land and water points considered for the sensitivity analysis are marked, positioned at $x = 100$, $y = 95$ for water and $x = 100$, $y = 105$ for land. The interface line runs along the west–east direction, reflecting the setup for simulating the diurnal cycle of a sea–land breeze developing in the south–north direction. The red box highlights the zoomed area in **(b)** around the points used for the sensitivity analysis; the neighbouring cells in the south–north direction used for the analysis are also shown in **(b)** around the selected points.

sitivity of the parameters in unexplored ranges is not known a priori. The perturbation percentage in this work has been chosen to avoid unphysical values, but it must be noted that the aim of the present work is to introduce and test ML-AMPSIT functionalities in a simplified case study, whereas a more detailed analysis would require more attention to the choice of the parameter space.

The output variable for which sensitivity is evaluated is the south–north horizontal component of the wind $v$ in the lowest 10 vertical levels at two different locations in the domain, one over land and one over water. These two locations are chosen to evaluate the effects of varying land parameters over two completely different surfaces and to assess how changes in land properties can also influence atmospheric fields over water. The locations are also strategically chosen to be near the interface between the land and water regions to better capture the dynamics of the sea–land breeze circulation, which is expected to be most pronounced near this boundary. For both locations, the average output from three adjacent cells in the south–north direction is analysed in order to increase the representativeness of the results. The central points of these two locations are $x = 100$, $y = 95$ and $x = 100$, $y = 105$ for water and land respectively, i.e. in the central cell in the west–east direction, with five grid points to the north and to the south of the land–water interface (Fig. 3).

## 4 Results

### 4.1 Sea breeze ensemble

Before analysing the ML-AMPSIT results, this section presents an overview of the output of the WRF/Noah-MP simulations, focusing on the horizontal south–north wind component $v$ at the two locations selected for the application of ML-AMPSIT. Figure 4 shows the ensemble time series of $v$ during the entire period analysed at the first vertical level. The daily cycle of the sea breeze is evident at both locations as the velocity changes sign according to the radiation pattern and the varying horizontal temperature and pressure gradient; i.e. $v$ is negative (northerly, from land to sea) during the night and positive (southerly, from sea to land) during the day. It is worth noting that, even if only land parameters have been considered in this work, the spread of the ensemble tends to be larger over water than over land, especially before sunrise. Indeed, changes in land parameters affect the thermal contrasts between land and water and, thus, the characteristics of the sea and land breeze, including their timing and strength. This highlights that changes in surface parameters can influence atmospheric variables more than just locally, especially when they affect the development of thermally driven circulations.

Figure 5 shows the ensembles of the vertical profiles of $v$ in the lowest 200 m, containing the lowest 10 vertical levels, at three different times, 07:00, 13:00, and 19:00 UTC, which are representative, respectively, of the maximum intensity of the northerly land breeze, the morning transition between northerly and southerly wind, and the maximum intensity of the southerly sea breeze at the water point. It can be noted that the northerly land breeze (Fig. 5a, b) is shallower than the southerly sea breeze (Fig. 5e, f). Moreover, the comparison between Fig. 5e and f highlights the stronger effect of friction over land, with a more pronounced decrease in the wind speed close to the surface.

The ensemble variance is small near the ground over land and increases with height at 07:00 and 19:00 UTC. Over the water point, the spread is larger and more uniform along the entire vertical profile, especially at 13:00 UTC and 19:00 UTC. During the transition (Fig. 5c and d), the ensemble spread is very small over land, with all the simulations showing very small $v$ values along the entire vertical profile, whereas a large spread is observed over water, suggesting that the variations in the surface parameters investigated significantly influence the timing of the transition from land to sea breeze over water, although preserving the shape of the vertical profile.

### 4.2 ML-AMPSIT output

After all the steps discussed in Sect. 2.1, the basic output that the ML-AMPSIT tool offers to the user is a composition of four plots similar to those reported in Fig. 6, which refers

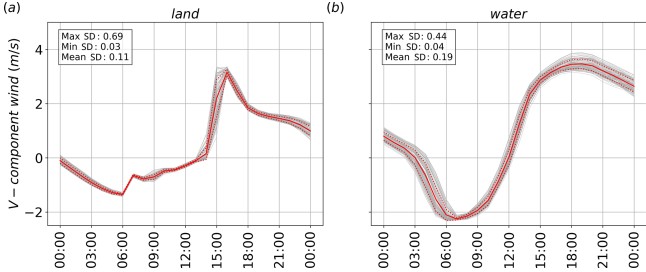

**Figure 4.** Time series of the south–north wind component at the first vertical level for the **(a)** land and **(b)** water points. The solid and dashed red lines represent the ensemble mean and standard deviation, respectively, while the grey lines represent the output of the single simulations. The values of the maximum, minimum and mean hourly standard deviations (SDs) are also reported.

to the results obtained for $v$ at the third vertical level over the water point (for this example, no averaging over multiple points was performed and all the simulation hours are shown in order to present the default output of the tool) us-
5 ing the LASSO algorithm. In particular, panel (a) shows the importance time series related to each of the six selected parameters; panel (b) shows the time evolution of the metrics, underlining how the correlation score and the errors eventually change over time; while panels (c) and (d) are specific
to a single hour selected by the user, showing, respectively, the goodness of fit and the ranking of the feature importance for that hour. In the following sections, the outputs from ML-AMPSIT will be aggregated to perform convergence analysis and a method comparison.

**4.3 Convergence analysis**

Figures 7 and 8 show the convergence of the MSE and of the feature importances computed by each method as a function of the number of simulations considered, i.e. the number of input–output relations used for training the algorithms. The
20 analysis of the convergence is important because it indicates when the regression tasks reach a stable state and when additional simulations do not significantly alter the results. When convergence is reached, it can be assumed that the obtained feature importance provides a reliable representation of the
25 underlying relationships in the system and that a sufficient number of simulations have been performed to capture the essential characteristics of the system under investigation.

For the sake of brevity, only results over the land region at the first vertical level are shown here, considering results
at 13:00 UTC for the feature importance and at four different times for the MSE convergence. However, the considerations reported below can be generalised to other times and to the water point since the methods maintain a similar speed of convergence during the entire run in the two analysed points.

Despite each method showing some differences, especially in the oscillations around the convergence values, four out of

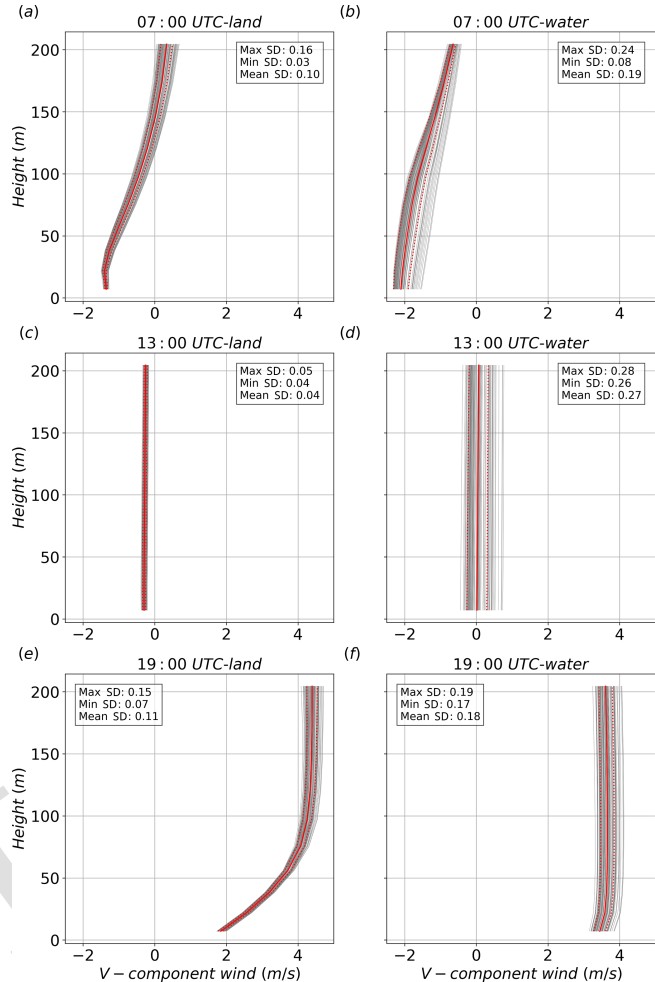

**Figure 5.** Vertical profiles of the south–north wind component in the lowest 200 m above the surface at **(a–b)** 07:00, **(c–d)** 13:00, and **(e–f)** 19:00 UTC for the land (left column) and water (right column) points. The solid and dashed red lines represent the ensemble mean and standard deviation, respectively, while the grey lines represent the output of the single simulations. The values of the maximum, minimum and mean hourly standard deviations (SDs) are also reported.

seven of the proposed methods are able to reach a reasonably stable result after approximately 20 simulations. BRR, GPR, LASSO, and SVM are the fastest and most stable algorithms with regard to reaching convergence. On the other 40 hand, the decision-tree-based methods have significant oscillations, even after 20 realisations, and the metrics show that they are less consistent than the other methods. However, the results highlight that all the methods propose a stable and consistent ranking of the parameters' importance after 45 80 simulations and, in most cases, even with a much lower number of simulations.

It can be seen from Fig. 8 that the sum of the total Sobol indices $\sum S_{\text{Ti}}$ generated by GPR for $N = 10$ is greater than 1, whereas this effect mostly disappears for $N \geq 20$. The val- 50

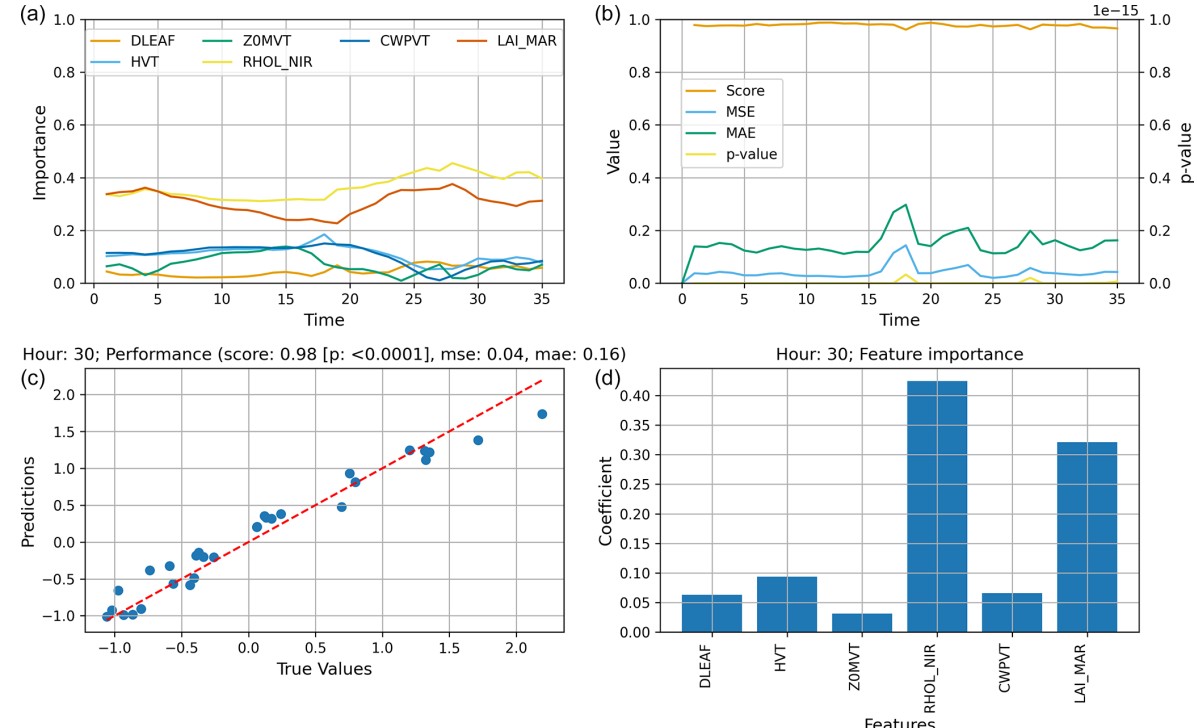

**Figure 6.** Example of the ML-AMPSIT output for a LASSO regression of 100 simulations with a duration of 36 h (spin-up phase included), focusing on $v$ at the third vertical level over the water point and displaying the metrics for the 30th simulated hour: **(a)** importance time series of the six parameters; **(b)** time evolution of the metrics $R^2$ (indicated as "score"), MSE, MAE, and $p$ value (testing the probability of unrelated variables to produce the same $R^2$); **(c)** quality of the test phase associated with the selected hour with the corresponding metrics; and **(d)** ranking of the importance of the features for the selected hour.

ues of the first-order terms $S_i$ obtained for $N = 10$ are indeed observed to be significantly different from $S_{Ti}$ because the higher-order terms in Eq. (4) are estimated to be relevant. For $N \geq 20$, the higher-order terms tend to become irrelevant, and so $S_i \approx S_{Ti}$, and $\sum S_{Ti}$ converges to 1. This occurs because, with a small number of simulations, the Sobol method tends to either overestimate or underestimate parameters' interactions due to insufficient sampling. Notably, BRR does not show the same overestimation of the indices even for $N = 10$.

## 4.4 Parameter importance analysis

Figures 9 and 10 show, respectively, the time series of the performance metrics for the south–north wind component at the lowest vertical level over the land and water regions, while Figs. 11 and 12 show, for the same variable and points, the time series of the feature importance for each of the proposed methods. It is important to underline that comparing the results of each surrogate model is the core of the ML-AMPSIT's robustness strategy. The agreement between the different models strengthens the reliability of the results and provides a form of self-validation.

In this example, GPR, BRR, LASSO, and SVM show the best metrics, suggesting that, in the proposed case study,

there is no relevant difference between non-linearity-aware approaches and linear approaches as they both correctly capture the relation between the tested parameters and the south–north wind component. These algorithms show very stable results, with slightly worse performance metrics occurring around 13:00 UTC over land and around 06:00 UTC over water. These times correspond to sudden changes in the ensemble spread (see Fig. 4), but the observed degradation in performance metrics is likely to be due to differences in the response of ensemble members to input variations rather than the time variation itself, which ML-AMPSIT cannot be aware of by design. The three decision-tree-based methods present a more irregular behaviour of the performance metrics, with higher errors and lower correlations. In particular, CART presents the worst performance metrics for this case study. The poorer metrics compared to more refined methods such as RF or XGBoost are expected since CART does not compute an ensemble of decision trees and does not consider the errors of the previous branches.

As shown in Figs. 11 and 12, all methods agree very well for both regions on the ranking and overall magnitude ratios of the feature importance, individuating similar patterns, with only minor differences; this is also considering the methods showing worse performance metrics (cf. Fig. 9).

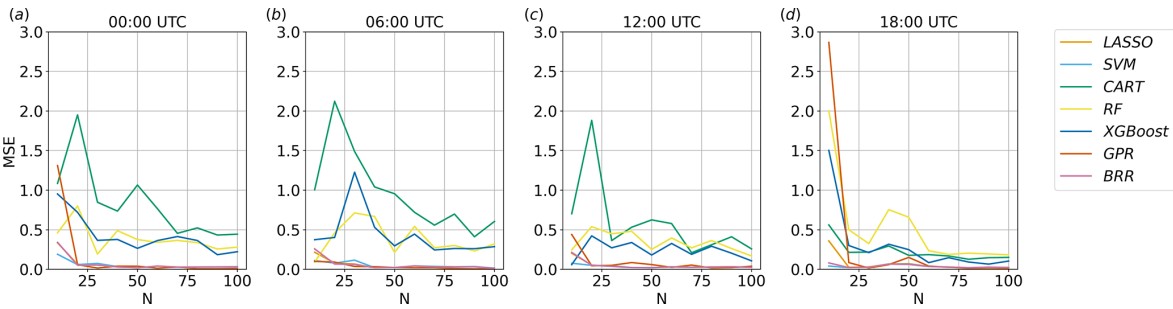

**Figure 7.** Convergence of the MSE with the number of realisations $N$ for each method implemented, considering $v$ over land at the first vertical level at four different times.

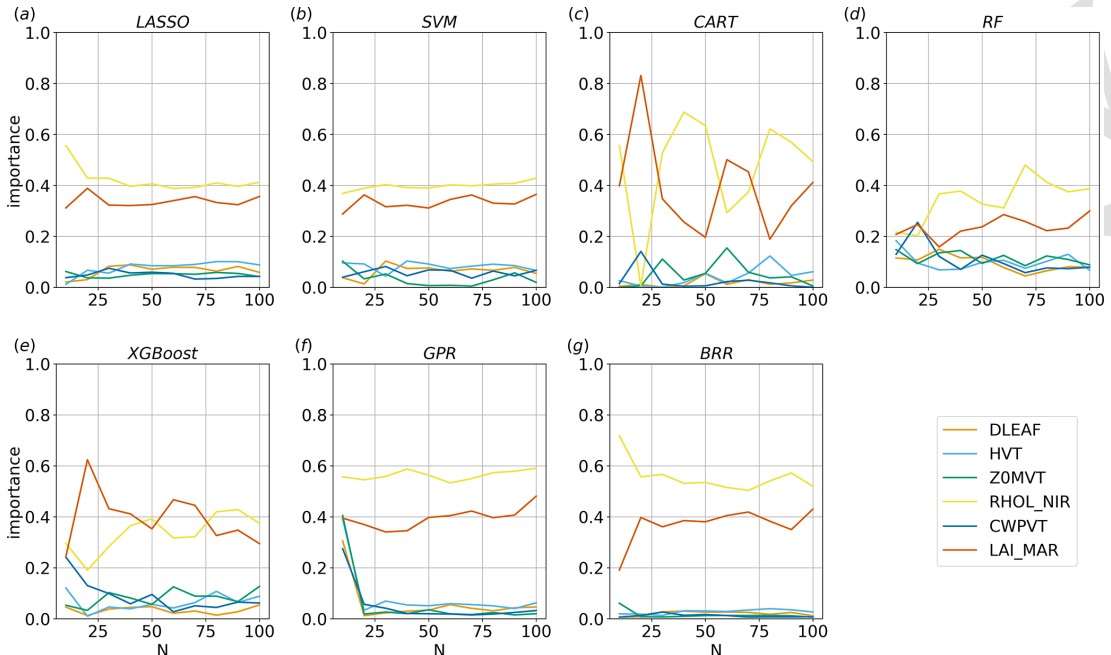

**Figure 8.** Convergence of the feature importance with the number of realisations $N$ for each method implemented, considering $v$ over land at the first vertical level at 13:00 UTC.

Figure 11 highlights cyclic trends of the parameters' importance over the land region, likely induced by the cycle of the diurnal thermally driven circulation. In particular, Z0MVT and RHOL_NIR alternate as the most important parameters, with RHOL_NIR dominating for most of the day, whereas Z0MVT becomes more important close to sunrise and sunset. The short time windows in which Z0MVT appears as the dominant parameter correspond to the phases in which the vertical wind profile over land showcases the most pronounced shear in the lowest layers, as shown in Fig. 5a and e. This seems to indicate a stronger role of surface friction in dictating ensemble variability when stronger winds are present (Z0MVT directly influences surface friction). LAI_MAR shows an importance almost comparable to RHOL_NIR during the day, especially with LASSO, SVM, and RF, whereas its importance is lower during the night. The

other parameters seem to be more relevant at night, with the exception of DLEAF, which is always non-relevant for every method implemented.

Comparisons between Figs. 11 and 12 show that the results are more uniform over water than over land. In particular, over water, the ranking of the parameters does not show significant variations throughout the whole day. The dominant parameters are RHOL_NIR and LAI_MAR, with Z0MVT always showing low importance values. Since the sea breeze is driven by thermal contrasts, it is expected that the parameters mainly affecting temperature, such as the reflectivity and the leaf area index, are also particularly significant for this case study. Among the selected parameters, RHOL_NIR plays a central role in the main radiative processes in Noah-MP, modulating the overall canopy albedo, defining the scattered fraction of leaf intercepted radiation, and ultimately

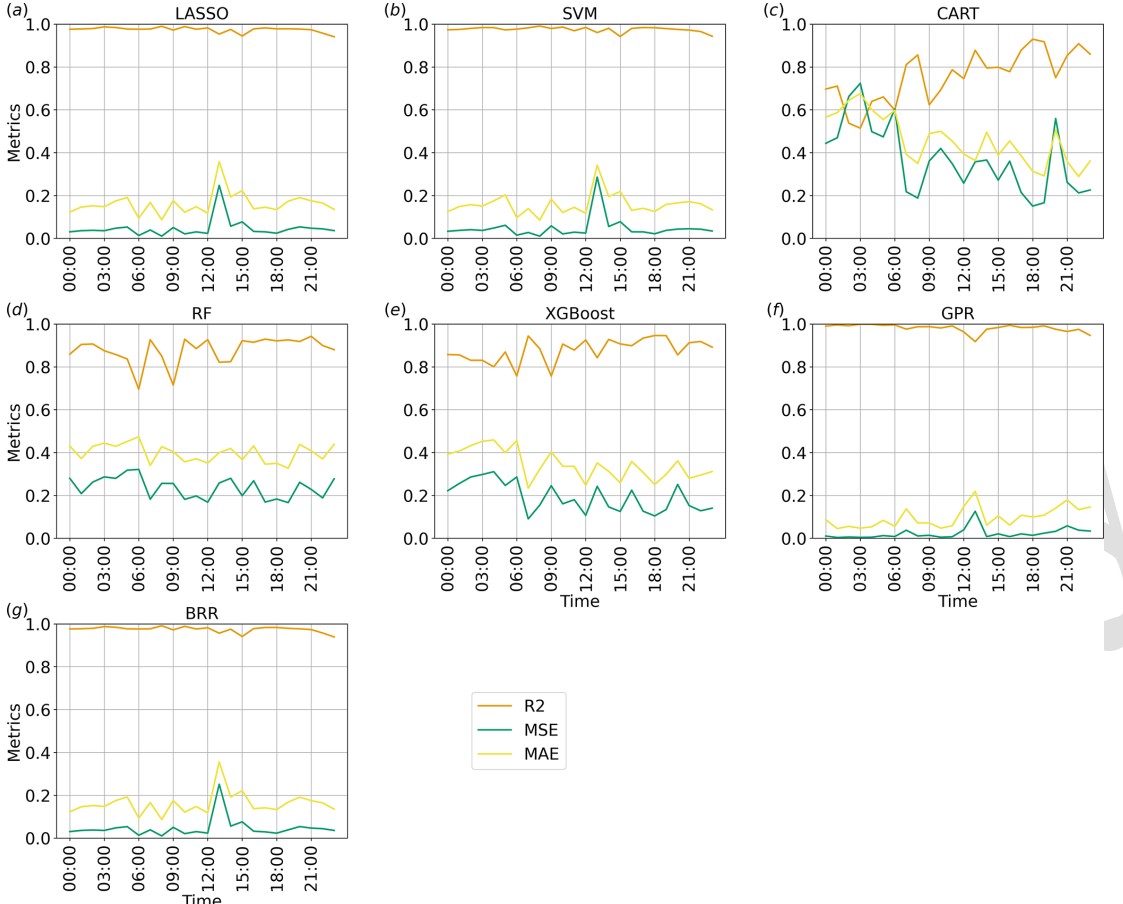

**Figure 9.** Time series of the performance metrics for each method implemented, considering $v$ over land at the first vertical level: green, red, and orange lines represent, respectively, $R^2$, MSE, and MAE.

entering the computation of all radiation fluxes. LAI_MAR is involved in important processes, such as determining the canopy gaps, the fraction of vegetation exposed to sunlight, and significantly affects both sensible and latent heat fluxes, as well as the leaf boundary resistance. Although HVT might be expected to be more important due to its influence on radiation and heat trapping, its importance is probably limited by the low canopy height in the selected grassland vegetation class. CWPVT, which enters the canopy wind extinction computation, and DLEAF, which mainly affects leaf boundary resistance, were expected to play a minor role in this setup with respect to the other parameters, mainly due to their secondary role in Noah-MP.

It is interesting to note that the decision-tree-based algorithms, CART, RF, and XGBoost, detect minor differences between the less relevant parameters overall, while the other methods, GPR, BRR, LASSO, and SVM, enhance the differences and define a clearer ranking in the first part of the day. The reason for these differences is reasonably due to the fact that, as mentioned in Sect. 2, the decision-tree-based algorithms are less strict about feature shrinkage compared to

other methods containing a regularisation term like LASSO, hence resulting in a less clear ranking in terms of feature importance with respect to the other methods. However, the relative importance between parameters is conserved overall; i.e. the feature importance ranking is mostly the same as in the other methods for the entire length of the simulation.

It is also worth noting that, considering the importance time series obtained from GPR and BRR surrogate models, the surrogate Sobol total index agrees very well with the feature importance scores of the other algorithms, which indicates that the Sobol indices derived from BRR and GPR and the feature importance derived from the other methods have equivalent sensitivity estimation capability when convergence is properly achieved.

## 4.5 Vertical variability

Figures 13 and 14 show, for the land and water regions, respectively, the variations in the feature importance in the lowest 10 vertical levels at different times. Since, as highlighted in the previous section, GPR is one of the methods presenting the best performance metrics, it has been chosen as the

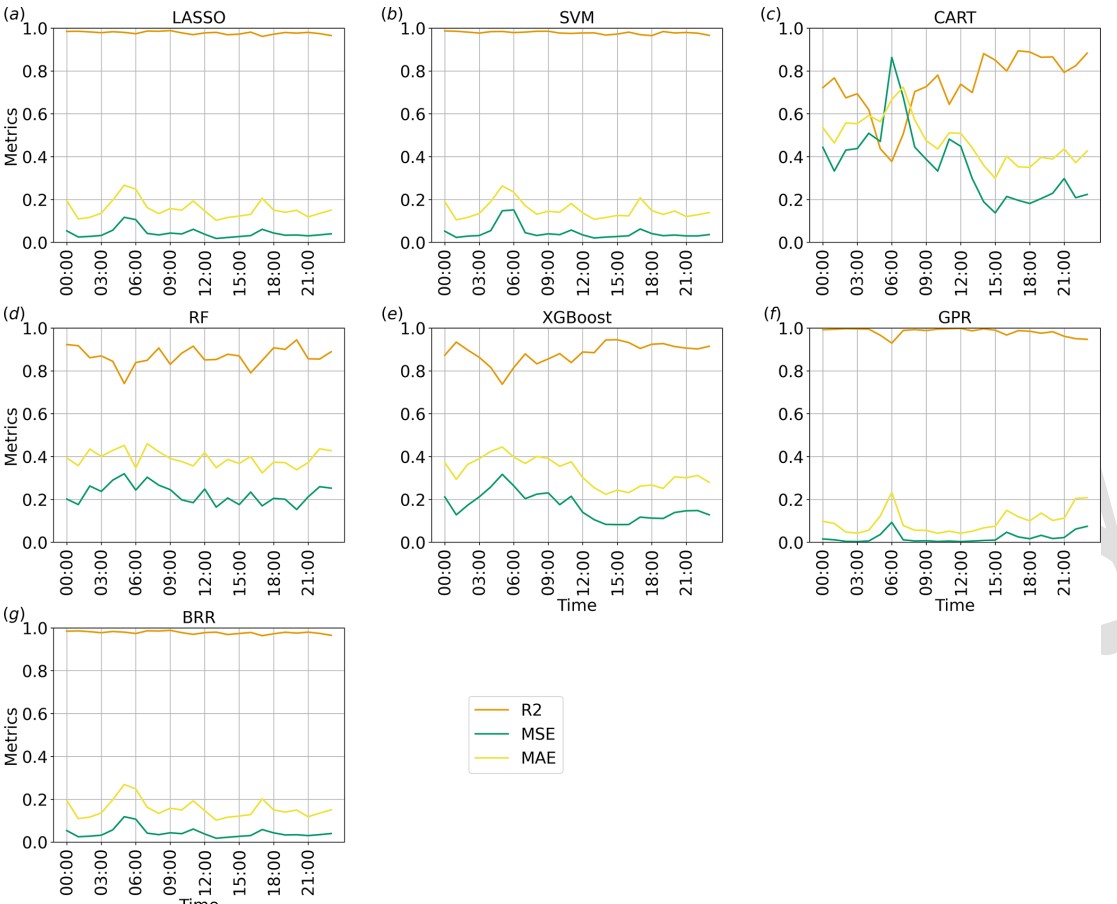

**Figure 10.** Time series of the performance metrics for each method implemented, considering $v$ over water at the first vertical level: green, red, and orange lines represent, respectively, $R^2$, MSE, and MAE.

reference algorithm for this analysis. However, the results obtained with GPR are consistent with those obtained with all the other methods, particularly with LASSO, SVM, and BRR.

Over the water region, the parameters' rankings do not show significant variations with height, except in correspondence with the nocturnal wind peak (06:00 UTC), when LAI_MAR becomes more important than RHOL_NIR above the eighth model level. The situation is more complex over the land region, with more significant variations in the parameter importance with height. In particular, it can be seen that Z0MVT is more important close to the surface, especially when wind speed is stronger (06:00 and 18:00 UTC), coherently with the results shown in Fig. 11, showing that friction affects the results, especially close to the surface. In contrast to the decreasing vertical importance of Z0MVT, the importance of LAI_MAR and RHOL_NIR tends to increase with height (Fig. 13). The importance ranking converges to the water region scenario shown in Fig. 14 above the lowest two vertical levels at 06:00 UTC and above the lowest five to six vertical levels at 18:00 UTC, i.e. above the layer in which

friction plays the most important role. On the other hand, when the wind speed is weak, i.e. at 00:00 and 12:00 UTC, the vertical profile of the parameters' importance values is similar over land and water at all the vertical levels investigated.

It is worth noting that the MSE for GPR, LASSO, BRR, and SVM does not show significant variations in the lowest 10 vertical levels, both over land and over water (Figs. 15 and 16), meaning that the observed variations in feature importance are related to changes in the input–output relation rather than to uncertainty issues. This is also supported by the fact that the metrics of these algorithms in Fig. 9 show no deterioration associated with the changes in feature importance shown in Fig. 11 and that these patterns are consistent across all the surrogate models. A slightly higher variability in MSE is shown by RF and XGBoost, whereas CART is the only method presenting a strong height dependence, particularly considering higher MSE values close to the surface at night over land and in correspondence with the northerly land breeze peak over water. These observations strengthen

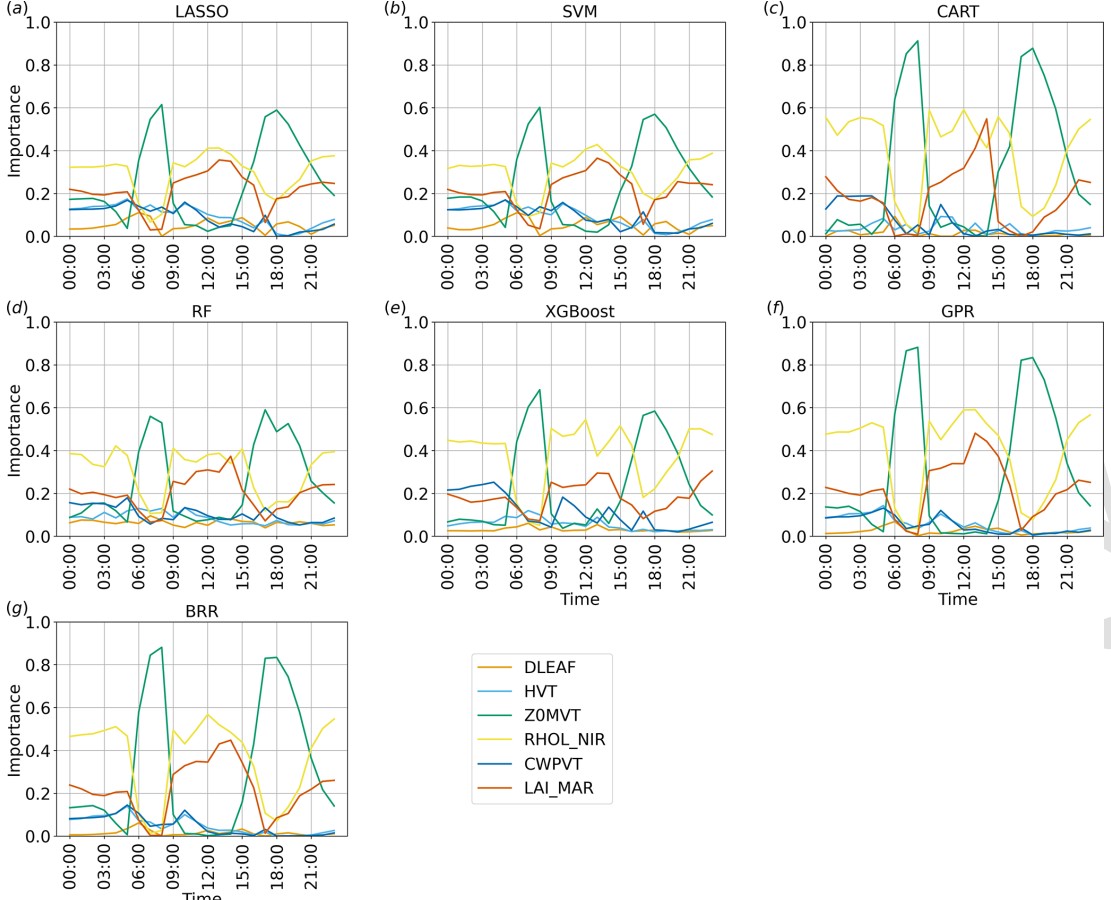

**Figure 11.** Time series of the importance of the parameters considered for each method implemented, considering $v$ over land at the first vertical level.

the evidence that this method is not performing well in this case study.

## 5   Discussion and conclusions

This paper presented a novel automated model parameter sensitivity and importance analysis tool (ML-AMPSIT) that applies different machine learning algorithms, namely LASSO, support vector machine, classification and regression trees, random forest, extreme gradient boosting, Gaussian process regression, and Bayesian ridge regression, to perform sensitivity analysis and to extract feature importance from input–output relationships. This tool was conceived to alleviate the computational burden usually associated with traditional global sensitivity analysis methods, which require a large number of model realisations, proposing an alternative approach using surrogate models or emulators. In fact, global sensitivity analysis methods, such as the Sobol method, demonstrate superior performance with respect to one-at-a-time approaches, which do not consider the interaction between parameters, but the large number of

model realisations needed often makes their use unfeasible for complex numerical models. On the other hand, surrogate models or emulators, trained using input–output pairs of the original high-fidelity model, offer a cost-effective means of generating accurate predictions of the output variable. The utilisation of machine learning techniques provides computationally efficient solutions while considering non-linearity and interactions between variables.

The advantage of implementing different methods, also within the same family of algorithms, is multifaceted. First, if different algorithms produce consistent results, this consistency increases the reliability and robustness of the outcome. Moreover, after assessing the consistency of the results between different models of the same family, it could be more convenient to rely on the fastest method instead of the most accurate. Second, the use of different families of algorithms extends the applicability and flexibility of the tool as their performance can vary in different scenarios.

The functionalities of the tool were tested and shown in a case study using the WRF meteorological model coupled with the Noah-MP land surface model. A sensitivity analysis applied to a set of Noah-MP parameters was presented for

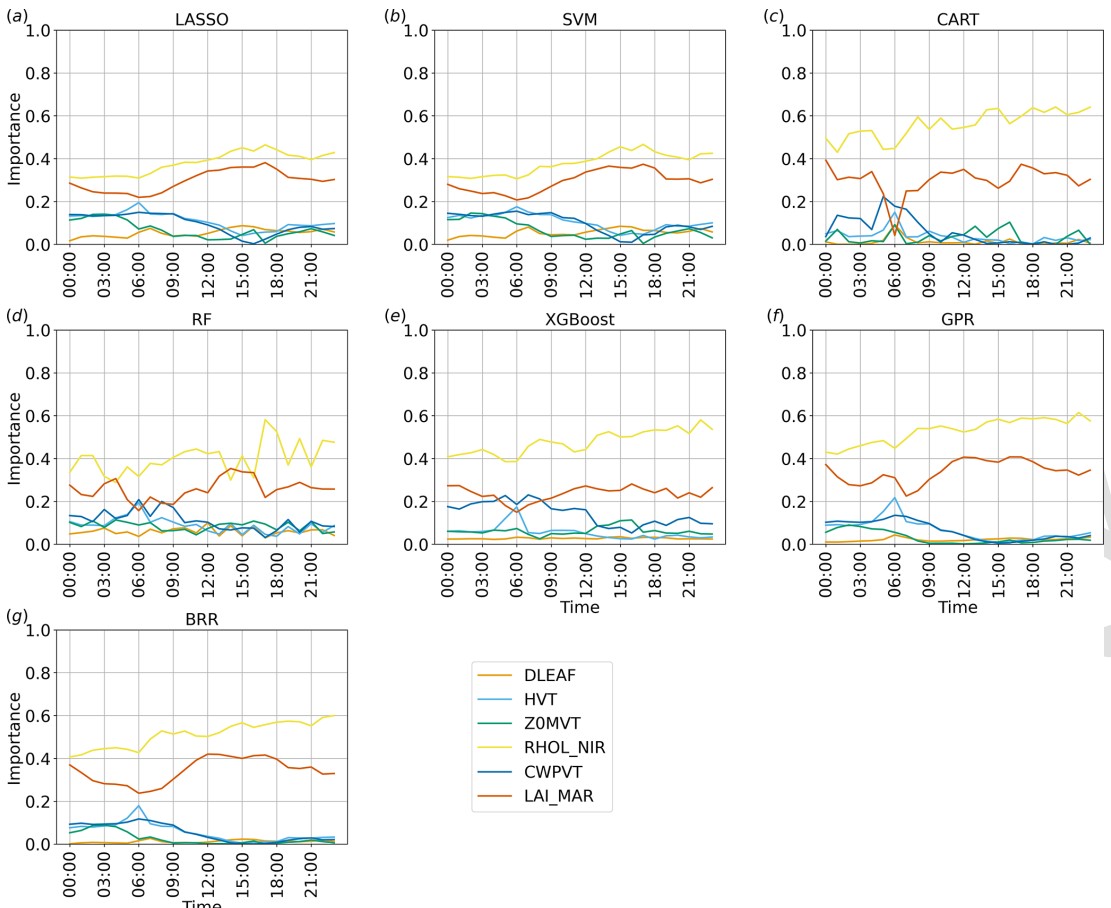

**Figure 12.** Time series of the importance of the parameters considered for each method implemented, considering $v$ over water at the first vertical level.

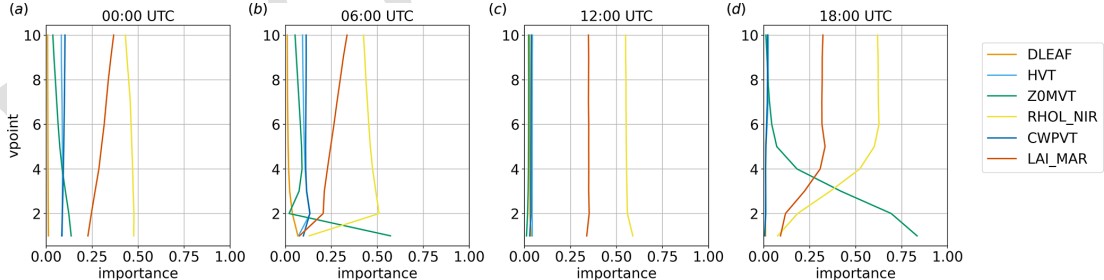

**Figure 13.** Parameter importance, considering $v$ in the lowest 10 vertical levels over the land region at different times, for the GPR method.

simulations of a sea breeze circulation over an idealised flat geometry. The different algorithms work as surrogate models of the original WRF/Noah-MP high-fidelity simulations and are able to accurately predict the original model behaviour and reach robust conclusions about the parameter sensitivity given a relatively small ensemble of model runs. The efficiency of the model emulation is also tested through the computation of Sobol total indexes from the training of Gaussian process regression and Bayesian ridge regression, with results being strongly consistent with those of the other pro-

posed feature extraction methods. By integrating multiple algorithms into a flexible framework, ML-AMPSIT offers a comprehensive and reliable approach for sensitivity analysis in complex models, also allowing the assessment of the uncertainty of the estimates by evaluating the spread between the outcomes of different algorithms.

Among the different methods, Gaussian process regression, LASSO, support vector machine, and Bayesian ridge regression emerged as the most reliable and robust. In contrast, decision-tree-based algorithms exhibited lower perfor-

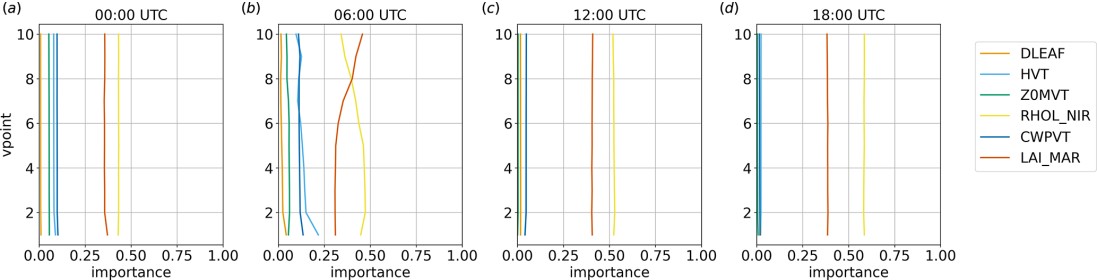

**Figure 14.** Parameter importance, considering $v$ in the lowest 10 vertical levels over the water region at different times, for the GPR method.

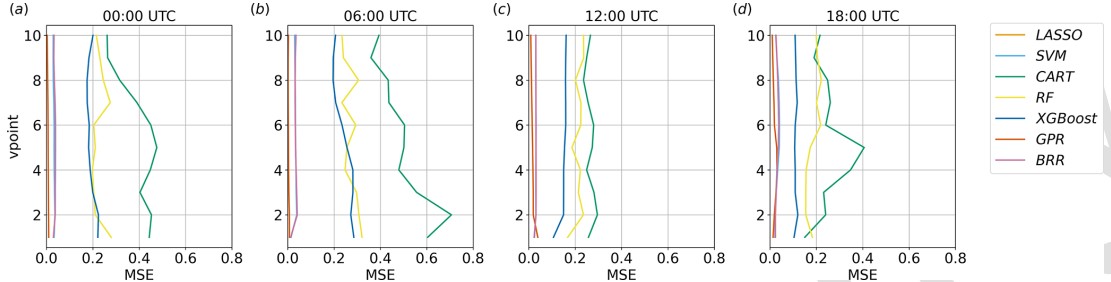

**Figure 15.** MSE for each method implemented, considering $v$ in the lowest 10 vertical levels over the land region at different times.

mance in terms of both the convergence with respect to the number of realisations and a higher uncertainty. In this case study, the linear models LASSO, support vector machine, and Bayesian ridge regression demonstrated equal perfor-
5 mance to the non-linearity-aware Gaussian process regression, suggesting the absence of strong non-linear relationships between the chosen parameters and the output variable in the analysed domain regions.

For the best algorithms, the convergence of the feature
importance was achieved with a small sample of about 20 simulations, whereas classical global sensitivity analysis approaches often require a much higher number of realisations. A qualitative comparison to evaluate the added value of ML-AMPSIT in terms of the number of simulations needed to
reach robust results can be performed considering two of the most advanced methods in global sensitivity analysis, i.e. the Morris method (Morris, 1991) and the Sobol method (Saltelli and Sobol', 1995), assuming the use of six parameters following a Latin hypercube sampling (Mckay et al., 1979) with
radial design (Campolongo et al., 2011). This sampling technique is one of the best trade-offs for decreasing the number of simulations needed compared to a full-factorial sampling (Saltelli et al., 2008). If parameter interactions are not relevant, such as for models with low complexity and low di-
mensionality, a viable strategy is to use the Morris method to find out the most and least relevant parameters. For $p$ points produced with a Latin hypercube sampling and for $k$ perturbations produced by radial design around these points, one perturbation for each input parameter, the total number
of model runs required, is $N = p(k+1)$. A sufficient num-

ber of points $p$ can be found in the literature, ranging from 10 to 50 (Campolongo et al., 2007), leading to 70–350 total simulations. However, even with this number of simulations, convergence is not guaranteed as it depends on the specific case. For more complex models, the Morris method
can be very inefficient in stating the true parameter relevance (it is usually considered to be only a proxy of the true sensitivities, depending on the number of interactions and non-linearities in the model; Cuntz et al., 2015). The Sobol method is able to weigh the interaction effects between each
parameter more accurately, but it is more demanding. Following Saltelli et al. (2010), to circumvent some constraints over the number of model runs required, the final number would be $N = p(k+1)(k+2)$, ~~which, using the previous assumptions for $p$, gives a minimum number of 560–2800~~
~~runs~~. Aside from the minimum amount computed above, real applications of the Sobol method can easily exceed this value to achieve robust results (Cuntz et al., 2015, 2016). However, this is not usually feasible for complex and computationally intensive models such as the WRF model.

It is then clear that ML-AMPSIT significantly reduces the number of simulations needed for sensitivity analysis and extraction of feature importance. Considering the fact that all the proposed regression methods in ML-AMPSIT intrinsically account for interactions between parameters, this high-
55 lights its added value over classical global sensitivity analysis methods and points out its possible applications, especially in cases when the use of classical global sensitivity analysis methods is not feasible. Furthermore, the intercomparison of

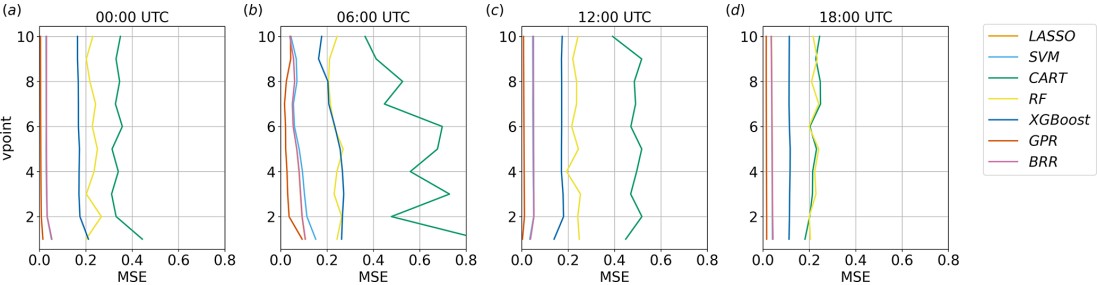

**Figure 16.** MSE for each method implemented, considering $v$ in the lowest 10 vertical levels over the water region at different times.

the results from different algorithms in ML-AMPSIT can reveal useful physical insights into model simulations.

It should be noted that the results presented in this paper are limited to the simple case study considered here to test the tool functionalities. In particular, it is expected that more simulations can be needed to train the algorithms in more complex scenarios, when non-linearities are more strongly involved in the input–output relations. However, while actual run times depend on the specific dataset and hardware, the speed improvements observed in our case study highlight the potential of ML-AMPSIT to enable large-scale sensitivity analysis and ensemble generation with significantly lower computational requirements. The generation of surrogate outputs was observed to be significantly faster than running high-fidelity WRF simulations, with run times being reduced from hours to seconds or minutes, depending on the algorithm. This efficiency enabled the generation of thousands of surrogate results that would not have been possible by relying solely on traditional simulations. Finally, it is worth noting that the application of the methods implemented in ML-AMPSIT is not only limited to the evaluation of land surface model parameters; these methods are inherently adaptable to any dataset containing input–output pairs, regardless of the data characteristics. This flexibility allows ML-AMPSIT to evaluate not only the influence of different input parameters but also the effects of different simulation setups, such as physical schemes, sub-processes, land cover, numerical strategies, or geometric configurations. By using data-driven modelling, these tasks can be accomplished more quickly and with potentially fewer data. Moreover, since input–output frameworks are ubiquitous in scientific and statistical domains, the reach of a data-driven tool like ML-AMPSIT potentially extends far beyond the specific examples mentioned here.

## Appendix A: ML-AMPSIT configuration file

```
{

  "comment1": "following lines are for generating realizations and populating the MPTABLES. folder is the reference simulation folder",
  "folder": "foldersim",
  "vegtype": 10,
  "totalsim": 100,
  "parameter_names": [ "DLEAF", "HVT", "Z0MVT", "RHOL_NIR", "CWPVT", "LAI_MAR"],

  "comment2": "for each parameters, the following matrix must contain: [middle point, percentage of perturbation]",
  "MATRIX": [
    [0.040, 50.000],
    [1.117, 50.000],
    [0.112, 50.000],
    [0.369, 50.000],
    [1.923, 50.000],
    [0.237, 50.000]
  ],

  "comment3": "following lines are specifications for ML-AMPSIT and post-processing",
  "input_pathname": "path/to/simulations/",
  "output_pathname": "path/to/output/",
  "totalhours": 36,
  "variables": ["V_MEAN","TH_MEAN"],
  "is_3d": [1,1],
  "regions": ["land", "water"],
  "verticalmax": 10,
  "tun_iter" : 10,

  "comment4": "following lines are for WRF post-processing: starting date, domain extension and points coordinate",
  "startTime": "2015-03-20 13:00:00",
  "ncfile_format":"wrfout_d01_2015-03-20_13_00_00",
  "ymax": 50,
  "xmax": 50,
  "y1": 30,
  "x1": 25,
  "y2": 20,
  "x2": 25

}
```

**Figure A1.** An example of the configuration file for the WRF/Noah-MP model case study.

## Appendix B: Abbreviations

| Abbreviation | Full form |
| --- | --- |
| ML-AMPSIT | Machine Learning-based Automated Multi-method Parameter Sensitivity and Importance analysis Tool |
| WRF | Weather Research and Forecasting |
| Noah-MP | Noah-Multiparameterization |
| ML | Machine learning |
| LASSO | Least absolute shrinkage and selection operator |
| SVM | Support vector machine |
| CART | Classification and regression trees |
| RF | Random forest |
| XGBoost | Extreme gradient boosting |
| GPR | Gaussian process regression |
| BRR | Bayesian ridge regression |
| GSA | Global sensitivity analysis |
| OAT | One-at-a-time |
| ARW | Advanced Research WRF |
| RRTM | Rapid radiative transfer model |
| YSU | Yonsei University |
| PBL | Planetary boundary layer |
| RBF | Radial basis function |
| MAE | Mean absolute error |
| MSE | Mean squared error |
| R2 | $R^2$ (coefficient of determination) |
| GUI | Graphical user interface |
| EE | Elementary effect |
| L1 | L1 regularisation (sum of absolute values) |
| L2 | L2 regularisation (sum of squares) |
| MPTABLE.TBL | Model parameter table |
| DLEAF | Characteristic leaf dimension |
| HVT | Height of vegetative canopy top |
| Z0MVT | Momentum roughness length |
| RHOL_NIR | Near-infrared leaf reflectance |
| CWPVT | Empirical canopy wind parameter |
| LAI_MAR | Leaf area index for March |

*Code and data availability.* The code of the ML-AMPSIT tool, along with detailed instructions on how to use it, is available at https://doi.org/10.5281/zenodo.10789930 (Di Santo, 2024b). The data extracted from the simulations and used for training the machine learning algorithms and producing the results presented in this paper are available at https://doi.org/10.5281/zenodo.14051616 (Di Santo, 2024a). This work used WRF version 4.4 (https://doi.org/10.5065/D6MK6B4K, Skamarock et al., 2019), which includes a built-in version of Noah-MP 4.4.

*Author contributions.* Conceptualisation was jointly performed by DDS, LG, CH, and FC. The methodology was developed by DDS, LG, and CH. DDS executed the formal analysis, developed the software, and produced the visual representations. The original draft was written by DDS, and editing and reviewing were performed by LG and CH. DDS and LG conducted the investigation process. LG acquired the financial support and was responsible for the project management and coordination. Supervision was carried out by LG, FC, and CH. Resources were provided by FC and CH.

*Competing interests.* The contact author has declared that none of the authors has any competing interests.

ther geographical representation in this paper. While Copernicus Publications makes every effort to include appropriate place names, the final responsibility lies with the authors.

*Acknowledgements.* We are grateful to Mathias W. Rotach for the fruitful discussions that inspired the approach used in this work.

*Financial support.* This research was funded by the Euregio Science Fund (3rd Call, IPN101) of the Europaregion Euregio.

*Review statement.* This paper was edited by Danilo Mello and reviewed by four anonymous referees.

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

**Remarks from the typesetter**

TS1    Please give an explanation of why this needs to be changed. We have to ask the handling editor for approval. Thanks.

TS2    Please give an explanation of why this needs to be changed. We have to ask the handling editor for approval. Thanks.