# Peer review of "ML-AMPSIT: Machine Learning-based Automated Multi-method Parameter Sensitivity and Importance analysis Tool"

_Geoscientific Model Development, 2024_

## Author Comment (AC2)

**RC2 Comments and responses**

We thank the Reviewer for her/his insightful comments. We appreciate the time and effort invested in providing detailed suggestions. Below, we address each comment in detail and outline the corresponding actions we have taken.

**General Comment:** A general suggestion to the manuscript readability due to the high number of acronyms and codes declared in the manuscript, is to add an acronym table that condensate abbreviations and, other table in the methodology section with the main characteristics of the mathematical techniques to help the reader to not to be overwhelmed with the immediate information of all these methods and their details.

**Reply:** We thank the Reviewer for this valuable suggestion. We agree that including such tables will improve the manuscript's readability and help readers better understand the methods used.

**Action:** We have added an acronym table to condense abbreviations and a table in the methodology section outlining the main characteristics of the mathematical techniques used in the study.

**General Comment:** Ending the Introduction to make smooth transitions a connector paragraph is needed to have smooth transitions between sections.

**Action:** We have added the following text to the end of the Introduction to ensure smooth transitions between sections:

> "This paper is organized as follows: Section 2 outlines the methodology used to develop ML-AMPSIT, including a detailed description of the machine learning models integrated into the tool and the workflow for performing sensitivity and importance analysis. Section 3 presents the case study involving the coupled WRF/Noah-MP model to demonstrate the application of ML-AMPSIT. The results of the sensitivity analysis are discussed in Section 4, highlighting the effectiveness of different machine learning models in identifying the key parameters for the case study presented in this paper. Finally, Section 5 concludes the paper with a summary of the findings and some insights into potential future work to further enhance the capabilities of ML-AMPSIT."

**General Comment:** Could be helpful as well an introductory paragraph of the Methods section.

**Action:** We have added the following text at the beginning of the Methods section:

> "In this section, we describe the methodological framework underlying this study. We begin with an overview of the ML-AMPSIT workflow, detailing the process from the selection of the input parameters to the sensitivity analysis phase. We then introduce the Sobol method, a variance-based technique used for GSA. Finally, we provide a description of the machine learning algorithms integrated into the tool, highlighting their main characteristics, how they are implemented and used in ML-AMPSIT and the rationale behind their selection."

**Specific Comment:** In Page 7, Eq. (2). Terms definition missing $V\{i,j,...\}$

**Action:** The definition of $V_{i,j,...}$ has been added:

> "where $V_i$ is the main effect variance, representing the contribution of the i-th input parameter to the output variance, $V_{ij}$ is the second-order interaction effect variance, representing the combined contribution of the i-th and j-th input parameters to the output variance, and so on up to $V_{12..k}$, which represents the interaction effect variance of all $k$ input parameters together."

**Specific Comment:** Page 12 Eq(5). Introduce terms in the equation that are not described in the text $\theta\_s$ and put units into []. What is TOP-MODEL?

**Reply:** We have defined $\theta_s$. TOPMODEL is a surface runoff model, we have added this information in the text, along with a reference. According to the journal's standard, we think that units should not be put into [] in the text.

**Action:** We have modified the text as follows:

> "where the surface potential temperature $\theta_s = 280$ K, $\Gamma = 3.2$ K km$^{-1}$, $\Delta\theta = 5$ K, and $\beta = 0.002$ m$^{-1}$."

"while the surface runoff parameterization TOPMODEL (Niu et al. 2007) with groundwater option is used for runoff and groundwater processes"

**Specific Comment:** Paragraph 355, page 13. Give more arguments about the selection of the two locations (one over land and one over water), I know these are very different locations but explain to the reader that you want to have two places that represent different dynamics in the model due to the input parametrizations of each site.

**Action:** We have added the following text to clarify the choice of the two locations:

"These two locations are chosen to evaluate the effects of varying land parameters over two completely different surfaces and to assess how changes in land properties can influence atmospheric fields also over water. The locations are also strategically chosen near the interface between the land and water regions to better capture the dynamics of the sea/land breeze circulation, which is expected to be most pronounced near this boundary."

**Specific Comment:** Section 3.2 Model setup should have a Figure with the characteristics of the domain or at least a table that summarizes the main characteristics of the model domains.

**Action:** We have added a figure to Section 3.2 that summarizes the main characteristics of the model domain.

**Specific Comment:** Agree with the minor comment on CC1: 'Comment on gmd-2024-56', Benjamin Püschel, 21 Jun 2024 :

"The quality of most figures is not entirely satisfying but could be improved with relatively little effort. For instance: Add a grid to the background of all figures. Increase font size in legends of Figs 3 & 4. Increase font size of labels in Fig 5 and title of subplot c). Add a second y-axis for the p-value in Figs 5, 8, 9 as it is close to 0. Swap x- and y-axis in Figs 12, 13, 14, 15 since height coordinates are usually represented on the y-axis. Increase line

width and use both colors and line styles to differentiate between lines in all plots. This would greatly increase visibility, especially for color-blind people. Is there a reason why the area under the curves is colored in the feature importance timeseries? (Figs 5, 10, 11)."

**Action:** All the suggestions have been considered and implemented into the manuscript.

**Specific Comment:** About the references section: It is suggested to add a couple references more from the year 2024 to update the state of the art of the manuscript. Put all the dates in the reference section homogeneously, i.e. all "....(year).....no: ".........(month year)..........."

**Action:** We have added some more recent references referring to the year 2024. In the revised manuscript the references are compliant with the GMD standards.

---

## Author Comment (AC3)

**RC1 Comments and responses**

We thank the Reviewer for her/his insightful comments. We appreciate the time and effort invested in providing detailed suggestions. Below, we address each comment in detail and outline the corresponding actions we have taken.

**Comment:** In the introduction it is mentioned that "ML techniques have gained traction in weather and climate modeling and observations [...] particularly in parameter optimization tasks like calibration", but I feel several relevant works exploring the use of emulators for tuning weather prediction and climate models, closely related to the long-term aims of the authors as far as I can interpret, are missing. I feel these should be cited. Here are a few examples. Daniel Williamson, Michael Goldstein, Lesley Allison, Adam Blaker, Peter Challenor, Laura Jackson, and Kuniko Yamazaki, "History matching for exploring and reducing climate model parameter space using observations and a large perturbed physics ensemble" (2013) Fleur Couvreux et al., "Process-Based Climate Model Development Harnessing Machine Learning: I. A Calibration Tool for Parameterization Improvement" (2020) Katherine Dagon, Benjamin M. Sanderson, Rosie A. Fisher, and David M. Lawrence, "A machine learning approach to emulation and biophysical parameter estimation with the Community Land Model, version 5" (2020) Duncan Watson-Parris, Andrew Williams, Lucia Deaconu, and Philip Stier, "Model calibration using ESEm v1.1.0 – an open, scalable Earth system emulator" (2021) Davide Cinquegrana, Alessandra Lucia Zollo, Myriam Montesarchio, and Edoardo Bucchignani, "A Metamodel-Based Optimization of Physical Parameters of High Resolution NWP ICON-LAM over Southern Italy" (2023)

**Reply:** We thank the Reviewer for providing these valuable references. They are indeed relevant to our work and help illustrate the broader context of using emulators for tuning weather prediction and climate models.

**Action:** We have added the suggested references in the Introduction section.

**Comment:** In Page 4 it is stated that "There is no upper limit for the number of parameters that can be analyzed", but of course the higher the dimensionality the harder the training of a surrogate can become, It would be useful to specify here how the number of simulations required scales with the number of parameters.

**Reply:** We fully agree with the Reviewer's comment. The exploration of

how the number of simulations required scales with the number of parameters is missing in this paper, as the main aim was to present the functionalities of the ML-AMPSIT tool using a simple idealized case study. However, it is difficult to evaluate a priori how the number of simulations needed to train the surrogate models scales with the number of parameters, since it can be dependent on the case study. We have added this consideration in the text.

**Action:** We have updated the text to emphasize the importance of dimensionality. The revised text now reads:

> "There is no upper limit for the number of parameters that can be analyzed, but it is worth noting that the sensitivity analysis could converge significantly more slowly in high-dimensional (i.e., with more parameters) problems. Moreover, the scalability with the number of parameters can highly depend on the case study considered."

**Comment:** In Page 7, Eq. (2), a definition of the terms $V_{i,j,\dots}$ is missing, and should be added.

**Reply:** Thanks for noting this.

**Action:** The definition of $V_{i,j,\dots}$ has been added:

> "where $V_i$ is the main effect variance, representing the contribution of the i-th input parameter to the output variance, $V_{ij}$ is the second-order interaction effect variance, representing the combined contribution of the i-th and j-th input parameters to the output variance, and so on up to $V_{12..k}$, which represents the interaction effect variance of all $k$ input parameters together."

**Comment:** In the Sections from 2.3.1 to 2.3.5 it is unclear how these different algorithms are used to compute an importance metric for the parameters. As far as I understood, the Sobol indices (the first-order one specifically) are computed only using Gaussian processes and Bayesian ridge regression. What is then precisely done when using the other ML algorithms explained? This explanation should be added to the manuscript.

**Reply:** We recognize that the original manuscript did not clearly convey how the feature importance is obtained for each of the surrogate models used.

**Action:** We have inserted a new section to clarify this process:

**Feature importance computation**

Each of the algorithms implemented in this study provides a method for calculating feature importance, albeit through different approaches. In principle, a single sensitivity method could be used to evaluate feature importance across all algorithms. However, some algorithms have built-in methods specifically designed to align with their inherent characteristics.

- Fitting Methods: LASSO and SVM derive feature importance from the model coefficients. In these linear models, the magnitude of the coefficients indicates the strength and direction of the relationship between each feature and the target variable. Specifically, in the `scikit-learn` library, this can be accessed through the `best_estimator_.coef_` attribute. Larger absolute values of these coefficients indicate greater importance.

- Tree-based algorithms: for CART, RF, and XGboost, feature importance is assessed using the Mean Decrease in Impurity (MDI) method. This method quantifies the contribution of each feature to the overall prediction accuracy by measuring how much each feature decreases the impurity of the splits in which it is involved. For RF and XGboost, the final value is obtained by averaging over all the trees in the ensemble. In `scikit-learn`, these contributions are accessible through the `feature_importances_` attribute. The MDI method is particularly effective because it directly measures the impact of each feature on the model's decision process, providing a clear indication of feature importance.

- Probabilistic methods: GPR and BRR do not have a built-in mechanism for directly assessing feature importance. Therefore, in this work, the Sobol method was used to infer feature importance. Once built and tested against the original model outputs, the GPR and BRR surrogate models can be used to perform a GSA in substitution of the original model. By using a surrogate model, the computational cost of running the original model for a large number of input combinations is avoided. Instead, the surrogate model can

be used to generate a large number of input combinations with significantly less computational time and evaluate their impact on the output. Over these samples, in ML-AMPSIT the Sobol sensitivity indices are computed following the definition proposed by Saltelli et al. (2008). The user can then compare the Sobol indices evaluated with both GPR and BRR, providing information on their robustness and reliability. In the proposed tool, after the algorithm generates the optimal surrogate model, it uses the Python library `SALib` to compute the Sobol first-order index as a score for the sensitivity importance of each parameter. The Sobol total index and Sobol second-order interaction term are available for users who wish to examine the presence of strong parameter interactions.

Despite the differences in the feature importance calculation approaches of the different algorithms, each method is applied to standardized, non-dimensional data and each feature importance set is scaled between [0,1]. This ensures that feature importance scores are comparable across models. The primary objective of all these methods is to quantify the sensitivity of the model output to changes in the input features. Consequently, the feature importance scores obtained from these different methods provide a well-posed comparison of parameter sensitivities. By evaluating and comparing these scores, it is possible to gain a comprehensive understanding of the relative importance of each feature across different modeling approaches, which increases the robustness of the results."

**Comment:** In Page 10, Section 2.3.6, the authors state that "GPR is a non-parametric method, i.e., it does not make assumptions about the functional form of the relationship between the input and output variables". The underlying assumptions on the functional form are contained in the chosen kernel, so there are in fact assumptions one has to make when using Gaussian processes. Maybe the authors here mean that there is no assumption of linearity with the chosen RBF kernel (as they specify later on)? Also, it seems that the authors do train the parameters of the kernel (e.g., lengthscale), so the adjective "non-parametric" may be confusing here.

**Reply:** We acknowledge that the text was creating unintentional ambiguity regarding the assumptions made by Gaussian Process Regression (GPR).

**Action:** We have revised the text to clarify this point. The following text has been added:

> "GPR is often described as a non-parametric method because it does not assume a specific functional form for the relationship between input and output variables. Instead, it models this relationship as a distribution over possible functions, allowing for flexibility in the shape of the regression curve. However, it is important to note that there are underlying assumptions about the functional form embedded in the chosen kernel. The kernel influences the shape and properties of the functions that the Gaussian process can learn."

**Comment:** In Page 10, Section 2.3.7, it should be specified what E and H in the equations mean in the context of the problem considered.

**Action:** We have added definitions for E and H:

> "Defining both a prior distribution $p(H)$ for the model parameters $H$ and a likelihood function $p(E|H)$ for the ingested data $E$, the BRR model computes the posterior distribution over functions $p(H|E)$ given the observed data through the use of Bayes' theorem $p(H|E) = \frac{p(E|H) \cdot p(H)}{p(E)}$, where $p(E) = \int p(E|H) \cdot p(H) \, dH$ is the marginal likelihood."

**Comment:** In page 11, Section 2.3.7, the authors state "The same procedure used for the GPR algorithm to leverage the probabilistic output for deriving feature importance coefficients is also implemented here to compute the Sobol first-order sensitivity index". I find confusing why the probabilistic nature of GPR or BRR is important for the calculation of the Sobol indices. In principle also 'deterministic' emulators like neural networks can be used to calculate Sobol indices. Can the authors comment on what they mean with this?

**Reply:** We acknowledge the confusion regarding the probabilistic nature of the algorithms chosen to implement the Sobol method. There is nothing inherently special about the probabilistic nature of GPR or BRR for calculating the Sobol indices. These algorithms were selected because they do not

have in-built methods for feature importance analysis compared to the other algorithms implemented in ML-AMPSIT.

One of the aims of the paper is to introduce a refined methodology for sensitivity analysis that addresses common issues in the literature, such as the simplistic assumption of linearity and the absence of interaction effects. Thus, we aimed to implement the Sobol method, an advanced sensitivity analysis technique historically considered too computationally expensive for numerical weather prediction models, to explore how quickly this method could be executed using surrogate models.

The Sobol method could, in principle, be used with all the surrogate models chosen in ML-AMPSIT. However, to provide a validation mechanism, other algorithms were implemented with their specific methodologies to evaluate feature importance. This approach allows for the production of a reliable ensemble and offers a metric for comparing the Sobol indices obtained.

**Action:** We have added the new section "Feature importance computation" in the revised manuscript which should clear doubts about the connection between the used surrogate models and the computation of feature importance.

**Comment:** In Page 13, the authors write "The spread of the ensemble tends to be larger over water than over land, especially before sunrise, indicating that the variation of the input parameters has a larger effect on v over water". Since most of the parameters varied were land-related parameters, I find this seemingly counterintuitive. Do the authors have a qualitative explanation for that?

**Reply:** The development and strength of sea and land breezes depend on the temperature contrasts between land and water. Therefore, it is reasonable that changes in the land parameters also affect atmospheric variables, in particular wind speed, over water, due to possible differences in the temperature contrasts between land and water and, as a consequence, in the timing and strength of the sea and land breezes. This is particularly true for the water point chosen for this study, which is close to the land/water interface. It is more difficult to understand why the ensemble spread is larger over water than over land. It may be connected to the higher friction over land, which dampens the variability induced by changes in the parameters' values. However, we prefer not to add speculations on this aspect in the text.

**Action:** We have added the following text to the manuscript to clarify

this point:

> "It is worth noting that, even if only land parameters have been considered in this work, the spread of the ensemble tends to be larger over water than over land, especially before sunrise. Indeed, changes in land parameters affect the thermal contrasts between land and water, and thus the characteristics of the sea and land breeze, including their timing and strength. This highlights that changes in surface parameters can influence atmospheric variables not only locally, especially when they affect the development of thermally-driven circulations."

**Comment:** From Page 14, when presenting the results the authors refer to the "importance" of the parameters, but no formula for this was given, especially in the context of LASSO, SVM, CART, RF, XGBoost. Please add a proper definition of it in the manuscript.

**Reply:** We thank the Reviewer for this comment that helped to improve the clarity of the manuscript. We have added the new section "Feature importance computation" that should clarify this aspect.

**Comment:** In the end, Page 26, the authors state "It is then clear that ML-AMPSIT significantly reduces the number of simulations needed for sensitivity analysis and extraction of feature importance". I find this a bit of a strong statement that should be mitigated. It is by no means clear that 20 or 30 simulations will be sufficient to train the emulators to reach faithful outputs. Specifically, as pointed out by the authors, the comparable performance of the investigated methods suggests the absence of strong non-linearities, which obviously renders the training of the methods more efficient. I expect that in presence of strong non-linearities the amount of training data will need to be increased, and so it remains a question as to whether this number will be systematically smaller than the other existing methods.

**Reply:** We recognize that the statement in the paper was unintentionally implying a generality that is not guaranteed for different setups and case studies.

**Action:** We have added the following text to mitigate the statement and provide appropriate context:

"It should be noted that the results presented in this paper are limited to the simple case study considered here to test the tool functionalities. In particular, it is expected that more simulations can be needed for training the algorithms in more complex scenarios, when non-linearities are more strongly involved in the input-output relations."

---

## Author Comment (AC4)

**CC1 Comments and responses**

We thank Benjamin Püschel, Isabella Winterer, Prof. Andreas Stohl and Dr. Stefano Serafin for their insightful comments. We appreciate that you have chosen our paper for your seminar course. Your detailed suggestions will be very useful to improve the quality of the paper. Thanks! Below, we address each comment in detail and outline the corresponding actions we have taken.

**Major Comment:** Sec. 2.3.1-2.3.7 & L80-82 While it is stated that the included ML methods are among the most commonly used, further justification is needed as to why exactly these seven methods are utilized. In particular, the utilization of tree-based methods requires explanation, as they demonstrate lower performance compared to other methods. Could they perform better or give additional insights in other cases? Otherwise, they might not be useful enough to be included in the tool.

**Reply:** The choice of the algorithms was primarily influenced by their prevalence in the literature, particularly in other earth science fields concerning landslide susceptibility, fire susceptibility, etc. Moreover, the appeal of these algorithms lies in their simplicity and speed, attributes that are not always guaranteed by more advanced ML-based algorithms. As stated in the text, the potential unpredictability of one or more factors affecting the performance of a specific algorithm usually necessitates a trial-and-error approach. This means that an algorithm that performs poorly in one scenario might perform well in another, underscoring the scenario-adaptability nature of the multi-method approach proposed in this study. While tree-based methods performed worse in this case study, another more complex case study was explored (to be the focus of a new paper), where non-linearities were stronger, and non-linear regressors such as tree-based methods performed much better than linear regressions.

**Action:** We have added the following text to enhance clarity about the algorithm selection:

In the Introduction:

> "These algorithms have been chosen for their simplicity and speed and to create an ensemble of state-of-the-art ML models each employing distinct methodologies, so as to improve the flexibility of the tool and its performance in different possible applications. This diversity allows for a robust method of self-validation or

self-falsification of the results through comparative analysis, enhancing the reliability of the findings by ensuring that consistent results are not an artifact of a single modeling approach"

In the conclusions:

"The advantage of implementing different methods, also within the same family of algorithms, is multifaceted. First, if different algorithms produce consistent results, this consistency increases the reliability and robustness of the outcome. Moreover, after assessing the consistency of the results between different models of the same family, it could be more convenient to rely on the fastest method instead of the most accurate. Second, the use of different families of algorithms extends the applicability and flexibility of the tool, as their performance can vary in different scenarios."

**Major Comment:** Sec. 2.3.1-2.3.5 We suggest a more detailed description of how feature importance is calculated/extracted for the methods LASSO, Support Vector Machine, Classification and Decision Trees, Random Forest, Extreme Gradient Boosting. We realized that the sum of the importances of all features does not equal 1 for all ML methods, suggesting that the feature importances are not normalized (e.g. Figs 10 & 11). However, non-normalized feature importances would not allow for direct comparisons of values between different ML methods (as done in e.g. L443-446). An explanation of the feature importance calculation would greatly clarify these ambiguities.

**Reply:** Thanks for the suggestion and for noting that in some cases the sum of the feature importance was not 1. Indeed, we recognize that the original manuscript did not clearly convey how feature importance is evaluated for each of the surrogate models used. Moreover, most feature importance methods result in normalized values except for SVM and LASSO, which are now normalized in the new manuscript version. This now allows for a direct comparison between the different ML algorithms.

**Action:** We have added the following section to enhance clarity about Feature importance computation:

**Feature importance computation**

Each of the algorithms implemented in this study provides a method for calculating feature importance, albeit through different approaches. In principle, a single sensitivity method could be used to evaluate feature importance across all algorithms. However, some algorithms have built-in methods specifically designed to align with their inherent characteristics.

- Fitting Methods: LASSO and SVM derive feature importance from the model coefficients. In these linear models, the magnitude of the coefficients indicates the strength and direction of the relationship between each feature and the target variable. Specifically, in the `scikit-learn` library, this can be accessed through the `best_estimator_.coef_` attribute. Larger absolute values of these coefficients indicate greater importance.

- Tree-based algorithms: for CART, RF, and XGboost, feature importance is assessed using the Mean Decrease in Impurity (MDI) method. This method quantifies the contribution of each feature to the overall prediction accuracy by measuring how much each feature decreases the impurity of the splits in which it is involved. For RF and XGboost, the final value is obtained by averaging over all the trees in the ensemble. In `scikit-learn`, these contributions are accessible through the `feature_importances_` attribute. The MDI method is particularly effective because it directly measures the impact of each feature on the model's decision process, providing a clear indication of feature importance.

- Probabilistic methods: GPR and BRR do not have a built-in mechanism for directly assessing feature importance. Therefore, in this work, the Sobol method was used to infer feature importance. Once built and tested against the original model outputs, the GPR and BRR surrogate models can be used to perform a GSA in substitution of the original model. By using a surrogate model, the computational cost of running the original model for a large number of input combinations is avoided. Instead, the surrogate model can be used to generate a large number of input combinations with significantly less computational time and evaluate their impact on the output. Over these samples, in ML-AMPSIT the Sobol sensitivity indices are computed following the definition proposed by Saltelli et al. (2008). The user can then compare the Sobol indices evaluated with both GPR and BRR, providing information on

their robustness and reliability. In the proposed tool, after the algorithm generates the optimal surrogate model, it uses the Python library `SALib` to compute the Sobol first-order index as a score for the sensitivity importance of each parameter. The Sobol total index and Sobol second-order interaction term are available for users who wish to examine the presence of strong parameter interactions.

Despite the differences in the feature importance calculation approaches of the different algorithms, each method is applied to standardized, non-dimensional data and each feature importance set is scaled between [0,1]. This ensures that feature importance scores are comparable across models. The primary objective of all these methods is to quantify the sensitivity of the model output to changes in the input features. Consequently, the feature importance scores obtained from these different methods provide a well-posed comparison of parameter sensitivities. By evaluating and comparing these scores, it is possible to gain a comprehensive understanding of the relative importance of each feature across different modeling approaches, which increases the robustness of the results."

**Major Comment:** Sec. 2.3.8 The algorithm depends on an initial guess of the plausible ranges of the hyperparameters/features whose importance is being estimated. The range boundaries of the six tested hyperparameters are not clearly justified in this work, and they do not seem to be adjustable by the user (in configAMPSIT.json). Likely, the feature importance estimate will be inaccurate if the initial parameter ranges are unrealistic. Some additional discussion of this aspect, and greater flexibility in the configuration of the algorithm, would be desirable.

**Reply:** Thanks for this comment. Indeed, in the old version of the manuscript we forgot to explicitly mention the range of variability of the parameters considered in the sensitivity analysis. This was only present in the configuration file in Figure A1 in the Appendix. The range of variation of the parameters is indeed a central topic in sensitivity analysis. In the simple idealized case study presented in this paper to show the functionalities of ML-AMPSIT, we decided to use maximum variations of 50% of the default parameter value, which we checked to be compatible with the natural variability of each parameter without generating unphysical situations.

ML-AMPSIT allows users to change the percentage of variation for each parameter and to use different percentages for each parameter. The provided

example of the file configAMPSIT.json shows the array defining the reference value and perturbation percentage for each parameter, both of which are required to be defined by the user.

**Action:** We have added the following statement to underline the importance of the parameters' ranges:

> "The final perturbed model parameter ensemble contains 100 samples, each with different parameter values based on the associated Sobol sequences. The input ensemble is generated by perturbing the parameters by up to 50% of their reference value in the look-up table MPTABLE.TBL. It should be clear that the results of a sensitivity analysis, regardless of the approach chosen, always depend on the range of exploration of the parameters, and that their transferability to arbitrary ranges of values is not guaranteed if the true sensitivity of the parameters in unexplored ranges is not known a priori. The perturbation percentage in this work has been chosen to avoid unphysical values, but it must be noted that the aim of the present work is to introduce and test ML-AMPSIT functionalities in a simplified case study, while a more detailed analysis would require more attention to the choice of the parameter space."

**Major Comment:** The paper is highly technical but lacks physical interpretation of the results. Physical explanations like the one given in lines 434-435 should be added also elsewhere. This would help the readers to better understand the usefulness of the tool in the concrete case presented.

**Reply:** The main objective of this paper is to present the new tool to the community for potential users, describing how it implements a sensitivity analysis methodology that accounts for commonly missing factors in the present literature, such as the non-linearity nature of the input-output response and the complex interactions between parameters in high-dimensional problems. Therefore, the key points of this study are oriented toward the implementation of advanced sensitivity analysis methods considered to be too computationally expensive for numerical weather prediction models, which potentially become fast and cheap through the use of surrogate models. The user-oriented nature of the tool required a comprehensive description of the workflow and the introduction of a minimum background concerning the implemented models, which covered most of the paper. We appreciate however

the suggestion to delve more into the physical interpretation of the results, which could be beneficial also for the above-mentioned main aims of the paper.

**Action:** We have inserted additional parts in the text to expand the physical interpretation of the results, e.g.:

"In particular, Z0MVT and RHOL_NIR alternate as the most important parameters, with RHOL_NIR dominating for most of the day, whereas Z0MVT becomes more important close to sunrise and sunset. The short time windows in which Z0MVT appears as the dominant parameter correspond to the phases in which the vertical wind profile over land showcases the most pronounced shear in the lowest layers, as shown in Figure 5a,e. This seems to indicate a stronger role of surface friction in dictating ensemble variability when stronger winds are present (Z0MVT directly influences surface friction)."

"Conversely to the decreasing vertical importance of Z0MVT, the importance of LAI_MAR and RHOL_NIR tends to increase with height (Figure 13). The vertical importance ranking converges to the water region scenario shown in Figure 14 above the lowest two vertical levels at 06:00 UTC and above the lowest 5-6 vertical levels at 18:00 UTC, i.e., above the height at which friction is playing the most important role. On the other hand, when the wind speed is weak, i.e., at 00:00 UTC and 12:00 UTC, the vertical profile of the parameters' importance values is similar over land and water at all the vertical levels investigated."

"The results are more uniform over water than over land, and the ranking of the parameters does not show significant variations during the whole day. In particular, the dominant parameters are RHOL_NIR and LAI_MAR, with Z0MVT always showing low importance values. Since the sea breeze is driven by thermal contrasts, it is expected that the parameters mainly affecting temperature, such as the reflectivity and the leaf area index, are also particularly significant for this case study. Among the selected parameters, RHOL_NIR plays a central role in the main radiative processes in Noah-MP, modulating the overall canopy albedo,

defining the scattered fraction of leaf intercepted radiation, and
ultimately entering the computation of all radiation fluxes. LAI is
involved in important processes, such as determining the canopy
gaps, the fraction of vegetation exposed to sunlight, and signifi-
cantly affects both sensible and latent heat fluxes, as well as the
leaf boundary resistance. Although HVT might be expected to be
more important due to its influence on radiation and heat trap-
ping, its importance is probably limited by the low canopy height
in the selected grassland vegetation class. CWPVT, which en-
ters the canopy wind extinction computation, and DLEAF, which
mainly affects leaf boundary resistance, were expected to play a
minor role in this setup with respect to the other parameters,
mainly due to their secondary role in Noah-MP."

**Minor Comment:** In the model setup, while other boundary conditions
are reported, the sea surface temperatures used are not.

**Action:** The sea surface temperature has been added to the model setup
description.

**Minor Comment:** Reduce the number of plots/subplots, especially if
they don't contain additional information. e.g. only show subplots with
interesting vertical variation of Figs 12 & 13; One plot showing the mean
vertical variation in MSE over land instead of Fig 14 & 15 would be enough
to visualize the takeaways in L460-465.

**Action:** We have reduced the number of subplots of the mentioned Fig-
ures to showcase 4 timestamps instead of 8, to convey only the main concepts.
Concerning other Figures, such as Figs 8-11, the repetitiveness of plots con-
taining the same information is still considered very important to underline
the benefits of a multi-method approach, which is one of the main aims of
this paper. The agreement between the different models strengthens the re-
liability of the results and provides a form of self-validation, which is at the
core of the ML-AMPSIT's robustness strategy.

**Minor Comment:** The quality of most figures is not entirely satisfying
but could be improved with relatively little effort. For instance:

- Add a grid to the background of all figures.

- Increase font size in legends of Figs 3 & 4.

- Increase font size of labels in Fig 5 and title of subplot c).

- Add a second y-axis for the p-value in Figs 5, 8, 9 as it is close to 0.

- Swap x- and y-axis in Figs 12, 13, 14, 15 since height coordinates are usually represented on the y-axis.

- Increase line width and use both colors and line styles to differentiate between lines in all plots. This would greatly increase visibility, especially for color-blind people.

- Is there a reason why the area under the curves is colored in the feature importance timeseries? (Figs 5, 10, 11).

**Reply:** Thanks for these very valuable suggestions. We have implemented all of them to improve the quality and readability of the figures.

**Action:** All suggested improvements have been implemented in the manuscript

**Minor Comment:** Typos in L123, 128, 151, 170, Fig 1: scriptnames should be *.ipynb instead of *.ipybn.

**Action:** The typo has been corrected.

**Minor Comment:** L432 Fig 11 should be linked.

**Action:** We have added a link to Fig 11.

**Minor Comment:** The paragraph L411-422 could link to Figs 8 & 9 more often for clarity and convenience of reading.

**Action:** The links to Figs 8 & 9 have been increased for clarity.

---

## Referee Report (RR1)

The authors of the manuscript "ML-AMPSIT: Machine Learning-based Automated Multi-method Parameter Sensitivity and Importance analysis Tool" have thoroughly addressed all the comments from the interactive session. The current version of the manuscript is now much clearer, especially concerning the importance of the parameters, which was one of the major points potentially causing confusion.

I thank the authors for their careful work, and can now recommend the manuscript for publication in its current version.

---

## Referee Report (RR2)

Review of "ML-AMPSIT: Machine Learing-based Automated Multi-method Parameter Sensitivity and Importance analysis Tool"

The manuscript details the workflow and application of the newly developed ML-AMPSIT. The tool provides an automated framework to conduct sensitivity and importance analysis for model parameters using seven different machine learning algorithms. It was developed for the WRF model but is applicable to all models dealing with model parameters. The aim of the manuscript is the introduction of the new tool and the description of its capabilities to support scientists conducting their sensitivity analysis with respect to model parameters. The tool was applied to the WRF model coupled with the NOAH-MP parameterization to analyze the dependence of the model results on the input parameters. With this, the capabilities of the ML-AMPSIT have been demonstrated.

The manuscript is well written and of good quality. All reviewer comments have been sufficiently addressed during the first review process. Also, the manuscript and the code fit well into the scope of GMD. However, I have stated a few comments below that remain unclear for me after reading the script. I recommend the publication of the manuscript after addressing my below posted comments.

General comments:

As this manuscript is a description of ML-AMPSIT, I would have expected a stronger discussion about the performance of the tool and the interpretation of the results. For example, the authors may address questions like: What is the runtime of the evaluation tool for the given test case for a single time series? Is the difference in the simulation results of the ensemble significant to evaluate feature importance or do they reflect the intrinsic uncertainty we have to expect in model simulations? Given WRF, which provides a large variety of parameterizations, is the investigation of model parameters a reasonable approach or may different parameterizations result in more ensemble spread and, thus, lead to more uncertainty? Also connected to the last point: Is the tool also applicable to other model uncertainties, e.g., the choice of different parameterizations or input data as land cover, SST, or emissions in the field of air quality. This may especially be important for the use of WRF, where it is more likely to first test different parameterization on their performance before evaluating the parameters within a single parameterization.

The idea of ML-AMPSIT is to construct surrogate models to evaluate the sensitivity of the model to certain parameters as well as the importance of these parameters. By performing sensitivity and importance analyses, the overarching goal is to improve the models performance. For me, it is not clear how the tool can support this. Model performance is usually evaluated against observations, which seem not to be included in the described approach. Do the surrogate models allow for testing further sets of model parameters to identify the set which best matches with the observations? Or does the surrogate model provide the best set of parameters by itself? In this case, is it recommended to increase the assumed uncertainty in model parameters to ensure that the surrogate models include all possible solutions for the prediction of model behavior for other choices of parameter values?

In the code repository, I'd recommend adding a readme file that details the workflow and basic principles of the ML-AMPSIT tool.

Minor comments:

- Line 57: ensemble perturbed parameter -> ensemble of perturbed parameters
- Line 67: Begin a new sentence "The study found…"
- Line 78: As I understand, the strength of ML-AMPSIT is its applicability to different models in the field of atmospheric research. I suggest adding this information to the sentence to stand out against the studies presented previously in the introduction.
- Line 108: the half-sentence "thanks to…" can be removed. It is a duplicate of the clause in line 105.
- Figure 1, step 4:
    - ML-AMPSIT.ipybn -> ML-AMPSIT.ipynb
    - looponfig.json -> loopconfig.json
- Figure 1, step 5:
    - ConvergenceAnlys.ipybn -> ConvergenceAnlys.ipynb (This file is missing in the code uploaded to zenodo. Please add.)
- Line 188: Please give more information about the p-value (I suppose from a significance test) in relation to $R^2$. How can it be interpreted and how is it related to $R^2$? What is the hypothesis to be tested?
- Line 225: What is the exact definition of $V_i$? Is it $V_i = VAR(Y(X_1, X_2, ..., X_i + \Delta, ..., X_k), Y(X_1, X_2, ..., X_i, ..., X_k))$? But if this is true, how is the perturbation $\Delta$ accounted for considering that (at least in the linear case) larger perturbations lead to larger effects in Y?
- In Eq. 4: Isn't summand $S_{13}$ missing according to the rule in Eq. 3?
- Line 261: "also known as a ridge-type regularization" can be removed as this is a duplicate of the statement in the previous sentence.
- Line 389: change the beginning of the sentence to "The initial atmospheric potential temperature …"
- Figure 3: I suggest decreasing the size of the figure but increase the font size of the text. I also suggest zooming in to the center of the figure to highlight the area of investigation. Also, the "3 adjacent grid cells", which are also considered in the analysis can be included in the plot. In the caption, the land area is denoted as "red area". At least in my copy it appears green. Please revise.
- Line 505: "increasing and others decreasing": What is increasing/decreasing? Please clarify.
- Line 507: The authors highlight the self-validating feature of the ML-AMPSIT tool, where an agreement between the different approaches is assumed to be a measure of robustness of the results. As 3 out of 7 algorithms perform worse than the others, the question arises at which point the authors claim the results as "not robust". Also keeping in mind that "worse performance" does not automatically mean "bad performance" in general. How can the user discriminate between robustness of the results and the results being unstable.
- Line 523: In this sentence, Fig. 12 can be referenced for clarity.
- Line 565: Add "Figure" before 11.
- The citation of He and Ek (2023) needs to be revised. The other co-authors do not appear in the reference.

---

## Author Response (AR3)

**Comments and responses**

We thank the Reviewer for her/his insightful comments. We appreciate the time and effort invested in providing detailed suggestions. Below, we address each comment in detail and outline the corresponding actions we have taken.

**1 General comments**

**Comment:** What is the runtime of the evaluation tool for the given test case for a single time series?

**Reply:** For the specific case study of this paper, ML-AMPSIT takes about 10 seconds to generate a single time series using any of the tree-based algorithms, up to about 1 minute for the slowest algorithms, i.e., BRR and GPR. It should also be noted that the original WRF simulations took 4 hours for each run, i.e. about 400 hours to build the 100 members of the ensemble, but the GPR surrogate model took about 1 minute to generate the 5000 WRF-surrogate outputs used to implement the Sobol method, which would have taken $4 \times 5000 = 20000$ hours using standard WRF simulations. However, these speed benchmarks are only guaranteed for the specific setup used in this study and the specificity of the hardware used to run these programs, and should not be considered a general speed benchmark. It would be expected that for larger datasets and more difficult-to-achieve hyperparameter tuning, the runtimes could increase with respect to the ones observed in this study. Even in such cases, the computational time is expected to be substantially lower than high-fidelity simulations.

**Action:** To highlight these aspects, text was added at lines 340-343: "The low computational cost of these emulators allowed us to employ a surrogate sampling generated by `sobol.sample()`, with 5000 input values (the user can change this value by modifying the configuration parameter *Nsobol* in *loopconfig.ipynb*) with the overall Sobol method calculations performed in minutes against a single traditional WRF simulation typically taking several hours.", and at lines 638-643: "However, while actual runtimes depend on the specific dataset and hardware, the speed improvements observed in our case study highlight the potential of ML-AMPSIT to enable large-scale sensitivity analysis and ensemble generation with significantly lower computational requirements. The generation of surrogate outputs was observed to be significantly faster than running high-fidelity WRF simulations, with runtimes reduced from hours to seconds or minutes, depending on the algorithm. This efficiency enabled the generation of thousands of surrogate results that would not have been possible by relying solely on traditional simulations."

**Comment:** Is the difference in the simulation results of the ensemble significant to evaluate feature importance or do they reflect the intrinsic uncertainty we have to expect in model simulations?

**Reply:** We thank the Reviewer for this pertinent question. If the ensemble spread were purely random or unrelated to the input parameters being tested, it would be unlikely for all algorithms to agree on parameter importance. In such a scenario, the regression task would either fail or overfit, resulting in unrealistic metrics and a lack of convergence despite an increase in sample size. However, the ability of our model to generate accurate predictions relative to simulated data suggests that most of the ensemble uncertainty is indeed attributable to variations in the selected parameters. This is highlighted in the text at lines 568-572: "It is worth noting that the MSE for GPR, LASSO, BRR and SVM does not show significant variations in the lowest 10 vertical levels both over land and over water (Figures 15 and 16), meaning that the observed variations in feature importance are related to changes in the input-output relation rather than to uncertainty issues. This is also supported by the fact that the metrics of these algorithms in Figure 9 show no deterioration associated with the changes in feature importance shown in Figure 11, and that these patterns are consistent across all the surrogate models."

**Comment:** Given WRF, which provides a large variety of parameterizations, is the investigation of model parameters a reasonable approach or may different parameterizations result in more ensemble spread and, thus, lead to more uncertainty? Also connected to the last point: Is the tool also applicable to other model uncertainties, e.g., the choice of different parameterizations or input data as land cover, SST, or emissions in the field of air quality. This may especially be important for the use of WRF, where it is more likely to first test different parameterization on their performance before evaluating the parameters within a single parameterization.

**Reply:** While this study focuses on a specific set of parameters within a particular land surface model, the ML-AMPSIT framework, in principle, is indeed adaptable to a variety of input-output scenarios. This flexibility means that ML-AMPSIT could be applied to different parameterization schemes, varied land cover types, alternative grid resolutions, and other simulation setups. Results of sensitivity analyses performed with ML-AMPSIT can also be compared to observations to evaluate the best model configuration.

**Action:** To emphasize this point, the following text has been added to the conclusions at lines 644-650: "Finally, it is worth noting that the application of the methods implemented in ML-AMPSIT is not only limited to the evaluation of land surface model parameters; these methods are inherently adaptable to any dataset containing input-output pairs, regardless of the data characteristics. This flexibility allows ML-AMPSIT to evaluate not only the influence of different input parameters, but also the effects of different simulation setups, such as physical schemes, subprocesses, land cover, numerical strategies, or geometric configurations. By using data-driven modelling, these tasks can be accomplished more quickly and with potentially less data. Moreover, since input-output frameworks are ubiquitous in scientific and statistical domains, the reach of a data-driven tool like ML-AMPSIT potentially extends far beyond the specific examples mentioned here."

**Comment:** By performing sensitivity and importance analyses, the overarching goal is to improve the models performance. For me, it is not clear how the tool can support this. Model performance is usually evaluated against observations, which seem not to be included in the described approach.
Do the surrogate models allow for testing further sets of model parameters to identify the set which best matches with the observations? Or does the surrogate model provide the best set of parameters by itself?
In this case, is it recommended to increase the assumed uncertainty in model parameters to ensure that the surrogate models include all possible solutions for the prediction of model behavior for other choices of parameter values?

**Reply:** The tool allows any set of parameters to be tested, and although the present paper is based on idealised simulations, the same can also be done with real-case simulations, thus allowing the comparison with observations. It is important to note, however, that the goal of the tool is to evaluate how much an input parameter affects an output variable, which is quite different from the task of finding the parameter values that best fit observations. For this reason, it finds the most important parameters that affect the variance in the data, but does not provide the best set of parameter values. Such an additional feature could in principle be implemented, perhaps in a future

version of ML-AMPSIT. However, once knowing which parameters cause most of the variance within a perturbed ensemble, the user can potentially concentrate on these parameters to improve model results. Indeed, knowing which parameters are most critical to the simulation output highlights which values should be estimated with more care to improve model results.

The user can arbitrarily change the uncertainty in the model parameters, but in this paper we chose to limit the ranges to realistic values to avoid unphysical situations.

**Action:** The main aim of ML-AMPSIT is summarised at lines 106-111: "ML-AMPSIT guides the user through the different steps of the sensitivity and importance analysis, allowing, on the one hand, for a simplification and automatisation of the process and, on the other hand, for extending the application of advanced sensitivity and importance analysis techniques to complex models, through the use of computationally inexpensive and non-linear interaction-aware methods. Once knowing which parameters cause most of the variance within a perturbed ensemble, the user can potentially concentrate on these parameters to improve model results. Indeed, knowing which parameters are most critical to the simulation output highlights which values should be estimated with more care to improve model results."

**Comment:** In the code repository, I'd recommend adding a readme file that details the workflow and basic principles of the ML-AMPSIT tool.

**Reply:** The README.md file in the code repository (https://dx.doi.org/10.5281/zenodo.10789930) explains the aims and principles of ML-AMPSIT and delves into the description of each file following a sequential order as intended in the workflow.

**2 Minor comments**

**Comment:** Line 57: ensemble perturbed parameter -> ensemble of perturbed parameters

**Action:** Text changed accordingly.

**Comment:** Line 67: Begin a new sentence "The study found..."

**Action:** Text changed accordingly.

**Comment:** Line 78: As I understand, the strength of ML-AMPSIT is

its applicability to different models in the field of atmospheric research. I suggest adding this information to the sentence to stand out against the studies presented previously in the introduction.

**Action:** We have clarified at lines 644-650 that ML-AMPSIT is designed to work with various input-output datasets, although this study focuses on land surface model parameters in WRF.

**Comment:** Line 108: the half-sentence "thanks to..." can be removed. It is a duplicate of the clause in line 105.

**Action:** Text changed accordingly.

**Comment:** Figure 1, step 4: ML-AMPSIT.ipybn $->$ ML-AMPSIT.ipynb ; looponfig.json $->$ loopconfig.json

**Action:** Typos in filenames corrected.

**Comment:** Figure 1, step 5: ConvergenceAnlys.ipybn $->$ ConvergenceAnlys.ipynb (This file is missing in the code uploaded to zenodo. Please add.)

**Reply:** The file ConvergenceAnlys.ipynb produces convergence plots (Figures 7 and 8 in the manuscript) from the file generated by ML-AMPSITloop.ipynb. However, its structure is not general enough to be used by any user without changes, because it depends on the details of the sensitivity analysis performed. Therefore, we decided not to include it in the main repository in the current version of the tool. However, users can easily perform convergence analyses, as presented in the manuscript, from the results provided by ML-AMPSIT, in particular by ML-AMPSITloop.ipynb. To avoid confusion, we have removed the convergence analysis step from Figure 1 and the workflow described in the paper. However, we have put the file ConvergenceAnlys.ipynb in the dataset repository (https://doi.org/10.5281/zenodo.14051616) as a reference for anyone interested in replicating the plots produced in the paper.

**Action:** The convergence analysis step has been removed from the workflow described in the paper and the file ConvergenceAnlys.ipynb has been put in the dataset repository (https://doi.org/10.5281/zenodo.14051616)

**Comment:** Line 188: Please give more information about the p-value (I suppose from a significance test) in relation to R2. How can it be interpreted and how is it related to R2? What is the hypothesis to be tested?

**Action:** Additional information about the p-value's relationship with $R^2$ has been added in the caption of Figure 6.

**Comment:** Line 225: What is the exact definition of Vi? Is it Vi = VAR(Y (X1, X2, ..., Xi + $\Delta$, ..., Xk), Y (X1, X2, ..., Xi, ..., Xk))? But if this is true, how is the perturbation $\Delta$ accounted for considering that (at least in the linear case) larger perturbations lead to larger effects in Y?

**Reply:** We thank the Reviewer for pointing out this. The perturbations $\Delta$ must be prescribed as random values that uniformly probe the output response in an arbitrarily wide range. In the present study, we chose the $\Delta$ ranges such that the values remain realistic to avoid unphysical output. Larger $\Delta$ values imply, at least in the linear case, larger effects on the model output, but this does not necessarily translate to larger parameter importance. However, it is true that the results of a sensitivity analysis, regardless of the approach chosen, always depend on the range of exploration of the parameters, and that their transferability to arbitrary ranges of values is not guaranteed if the true sensitivity of the parameters in unexplored ranges is not known a priori.

**Action:** The effect of varying the range of variability of the parameters is commented on at lines 426-431: "It should be clear that the results of a sensitivity analysis, regardless of the approach chosen, always depend on the range of exploration of the parameters, and that their transferability to arbitrary ranges of values is not guaranteed if the true sensitivity of the parameters in unexplored ranges is not known a priori. The perturbation percentage in this work has been chosen to avoid unphysical values, but it must be noted that the aim of the present work is to introduce and test ML-AMPSIT functionalities in a simplified case study, whereas a more detailed analysis would require more attention to the choice of the parameter space."

**Comment:** In Eq. 4: Isn't summand $S_{13}$ missing according to the rule in Eq. 3?

**Reply:** Yes, thanks for pointing out this error.

**Action:** Added the missing $S_{13}$ term.

**Comment:** Line 261: "also known as a ridge-type regularization" can be removed as this is a duplicate of the statement in the previous sentence.

**Action:** Revised as suggested.

**Comment:** Line 389: change the beginning of the sentence to "The initial atmospheric potential temperature ..."

**Action:** Text changed accordingly.

**Comment:** Figure 3: I suggest decreasing the size of the figure but increase the font size of the text. I also suggest zooming in to the center of the figure to highlight the area of investigation. Also, the "3 adjacent grid cells", which are also considered in the analysis can be included in the plot. In the caption, the land area is denoted as "red area". At least in my copy it appears green. Please revise.

**Action:** Figure 3 was revised according to the Reviewer's suggestions, highlighting the area of investigation and adjusting colours.

**Comment:** Line 505: "increasing and others decreasing": What is increasing/decreasing? Please clarify.

**Action:** Text changed to improve clarity: "Around these times, individual ensemble members exhibit divergent behaviour, some showing increases and some decreases in wind speed, which can complicate the prediction for the regression models".

**Comment:** Line 507: The authors highlight the self-validating feature of the ML-AMPSIT tool, where an agreement between the different approaches is assumed to be a measure of robustness of the results. As 3 out of 7 algorithms perform worse than the others, the question arises at which point the authors claim the results as "not robust". Also keeping in mind that "worse performance" does not automatically mean "bad performance" in general. How can the user discriminate between robustness of the results and the results being unstable.

**Reply:** The Reviewer is right that "worse performance" is different than "bad performance". The performance of the algorithms can be first assessed from the values of the evaluation metrics, as stated at lines 183-188. Then, the comparison between the results of the different algorithms helps to evaluate if a worse performance is also a bad performance or not. In the paper, although the metrics of the tree-based algorithms are quite lower than those of the other algorithms used, the robustness and stability of the results are inferred from the agreement among all the algorithms on the importance and ranking of the parameters, as well as from the convergence analysis of the results. This is stated, for example, at lines 485-490. Therefore, the user should

combine the information coming from the evaluation metrics and the comparison between the output of the different models to have a complete idea of their performance and thus to discriminate between "worse" and "bad" performance. The importance of comparing the results of the different models is highlighted at lines 500-502.

**Comment:** Line 523: In this sentence, Fig. 12 can be referenced for clarity.
**Action:** Reference to Fig. 12 added.

**Comment:** Line 565: Add "Figure" before 11.
**Action:** Text changed accordingly.

**Comment:** The citation of He and Ek (2023) needs to be revised. The other co-authors do not appear in the reference.
**Action:** The citation was revised.